# DialSim: A Real-Time Simulator for Evaluating Long-Term Multi-Party Dialogue Understanding of Conversational Agents

## Abstract

Recent advancements in Large Language Models (LLMs) have significantly enhanced the capabilities of conversational agents, making them applicable to various fields (*e.g.*, education). Despite their progress, the evaluation of the agents often overlooks the complexities of real-world conversations, such as real-time interactions, multi-party dialogues, and extended contextual dependencies. To bridge this gap, we introduce `DialSim`, a real-time dialogue simulator. In this simulator, an agent is assigned the role of a character from popular TV shows, requiring it to respond to spontaneous questions using past dialogue information and to distinguish between known and unknown information. Key features of `DialSim` include evaluating the agent's ability to respond within a reasonable time limit, handling long-term multi-party dialogues, and testing the agent's performance under randomized questioning with a diverse and high-quality question-answer dataset. We utilized this simulator to evaluate the latest conversational agents and analyze their limitations. Our experiments highlight both the strengths and weaknesses of these agents, providing valuable insights for future improvements in the field of conversational AI. `DialSim` is available at https://anonymous.4open.science/r/Simulator-A861.

## 1 Introduction

Recent advancements in Natural Language Generation (NLG) within Large Language Models (LLMs) have significantly enhanced the capabilities of conversational agents. These agents are now integral to various fields, including entertainment (Zhou et al., 2023; Chen et al., 2024) and education (Ait Baha et al., 2023; Waisberg et al., 2024), providing personalized interactions that cater to individual preferences and interests. As these agents continue to evolve and become more widely adopted, it is crucial to rigorously assess their performance in real-world scenarios to ensure they meet user expectations and function effectively.

Traditionally, the evaluation of conversational agents has relied on qualitative assessments of their responses. This process typically involves human evaluators or LLMs judging the quality of an agent's utterances (Adiwardana et al., 2020; Zhang et al., 2020; Roller et al., 2021; Shuster et al., 2022; Lee et al., 2023; Kim et al., 2024) or comparing responses between different agents on platforms like *Chatbot Arena* (Chiang et al., 2024). While these methods provide valuable insights into aspects such as naturalness and alignment with user instructions, they do not fully capture the complexities of real-world interactions.

In real-world dialogues, conversational agents encounter a range of challenges: engaging in real-time interactions, managing conversations with multiple participants, and recalling information from past dialogues. Therefore, a more comprehensive evaluation method is needed—one that accurately assesses an agent's ability to respond within a reasonable time limit, thoroughly understand multi-party dialogue contexts, and reason across previous interactions, reflecting the complexities of real-world conversations.

To address this, we propose `DialSim`, a dialogue simulator for real-time evaluation of a conversational agent's long-term multi-party dialogue understanding. As illustrated in Figure 1, `DialSim` places an agent in the role of the main character of a TV show, engaging it in an extensive conversation based on the show's scripted content. During this conversation, the agent is evaluated for its ability to

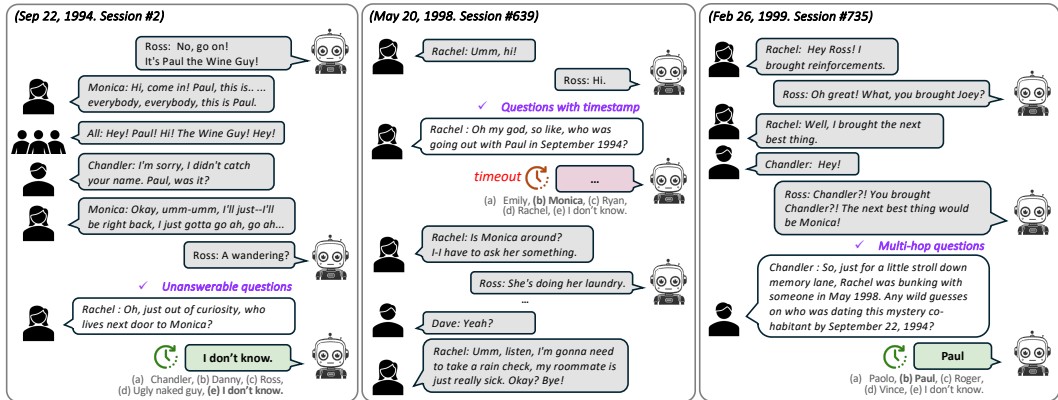

Figure 1: An overall process of `DialSim`. Gray speech bubbles indicate predetermined utterances from the script, and white speech bubbles indicate spontaneous questions asked during the simulation. Colored speech bubbles indicate the agent's responses to the questions. (Left) An unanswerable question. (Center) A question that references a specific time. (Right) A multi-hop question that requires understanding past sessions (*i.e.*, the Left and Center boxes). The question is asked in the format chosen by the user, either in a multiple-choice format or as an open-ended question.

respond appropriately to spontaneous questions from other speakers within a predefined time limit (*e.g.*, 1s / 3s / 5s). The agent should be able to answer questions based on the information from the past dialogue and acknowledge when it does not know the answer.

`DialSim` simulates three different dialogue environments based on scripts from popular TV shows (*i.e.*, Friends, The Big Bang Theory, and The Office), and it has the following four main features.

- **Time-Constrained Evaluation**: For a conversational agent to function effectively in real-time, the agent must be capable of updating its memory and generating responses on the fly. To evaluate this, `DialSim` measures the accuracy of responses within a predefined time limit. To the best of our knowledge, this is the first work that evaluates the performance of a conversational agent in a time-constrained environment.

- **Extensive Long-Term Multi-Party Dialogue**: `DialSim` simulates multi-party dialogues averaging 350k tokens in length, making it the longest dataset among existing long-term dialogue datasets. Throughout the simulation, the agent encounters complex questions that require comprehension and reasoning across several multi-party sessions, ensuring a thorough assessment of its long-term dialogue understanding capabilities.

- **Diverse and High-Quality Question-Answer Dataset**: We generated an average of 1,000 unique questions per session using two methods. First, we collected and refined questions from a fan quiz website that covers key events in each TV show. Second, we extracted temporal knowledge graphs for each session to formulate complex, multi-hop questions based on these graphs. We employed ChatGPT-4 (OpenAI, 2023b) to refine the fan quizzes and extract the knowledge graphs, ensuring high quality through manual review by the authors.

- **Randomized Questioning**: `DialSim` features an average of 1,300 conversation sessions occurring over a period of five years, as depicted in the corresponding TV show. For each session, a randomly selected character asks a question that is randomly sampled from an average of 1,000 candidates, at a random time. This setup allows us to test whether an agent can respond appropriately in a challenging and unpredictable environment, suitable for rigorous evaluation of conversational agents. Additionally, since a random set of questions is given for each test, repeating the tests multiple times allows us to measure the agent's performance variability and reliability.

## 2 RELATED WORKS

**Long-Term Dialogue Datasets**   Widely used dialogue datasets include DailyDialog (Li et al., 2017), which features everyday conversations; PersonaChat (Zhang et al., 2018), which contains

Table 1: Comparison of `DialSim` with other long-term dialogue datasets.

| Dataset | # of Turns[†] | # of Sessions[†] | # of Tokens[†] | Collection | Avg. # of Speakers | Real-Time Evaluation |
|---|---|---|---|---|---|---|
| MSC (train; 1-4 sessions) | 53.3 | 4 | 1225.9 | Crowdsourcing | 2.0 | ✗ |
| MSC (valid + test; 1-5 sessions) | 61.7 | 5 | 1670.9 | Crowdsourcing | 2.0 | ✗ |
| Conversation Chronicles | 58.5 | 5 | 1054.7 | LLM | 2.0 | ✗ |
| LoCoMo | 304.9 | 19.3 | 9209.2 | Crowdsourcing + LLM | 2.0 | ✗ |
| **DialSim (ours)** | **18986.3** | **1313.3** | **351996.3** | **Scripts** | **3.4** | ✓ |

†: averaged over dialogues

the persona of each character; Empathetic Dialogues (Rashkin et al., 2019), designed for emotional conversations. However, conversational agents built with these datasets were limited to single-session interactions. To address this, Multi Session Chat (Xu et al., 2022) was created, featuring up to five sessions per dialogue. Despite this improvement, generating longer dialogues via crowdsourcing remained challenging. To overcome this challenge, Conversation Chronicles (Jang et al., 2023) was developed by leveraging an LLM. Furthermore, LoCoMo (Maharana et al., 2024) evaluates dialogue comprehension of an agent through various tasks (*e.g.*, event summarization) in long-term dialogues. In contrast to other datasets generated through crowdsourcing or LLMs, the dialogues in `DialSim` are derived from TV show scripts, offering a unique set of benefits. These scripts feature multiple characters engaged in extended dialogues spanning several years, with character details (*e.g.*, relationships) evolving over time. Leveraging these characteristics, we developed a dialogue simulator that replicates extremely long-term dialogues with evolving relationships. Table 1 provides a detailed comparison between `DialSim` and other long-term dialogue datasets.

**Datasets Based on the Scripts of TV Shows** While both TV show scripts and other dialogue datasets effectively capture dialogue characteristics, scripts offer a significant advantage due to their abundance and accessibility. This makes them particularly valuable for various dialogue understanding tasks such as question answering (QA) (Yang & Choi, 2019; Sang et al., 2022), coreference resolution (Chen & Choi, 2016; Chen et al., 2017; Zhou & Choi, 2018), relation extraction (Rashid & Blanco, 2018; Yu et al., 2020), and summarization (Gorinski & Lapata, 2015; Papalampidi et al., 2020; Chen et al., 2022). Notable datasets derived from scripts include FriendsQA (Yang & Choi, 2019) and TVShowGuess (Sang et al., 2022). FriendsQA treats each scene in every episode as an independent conversation. Each question inquires about information related to the conversation, and the task is to find the corresponding answer spans. TVShowGuess is a multiple-choice QA dataset consisting of five different popular TV shows. The task is to identify anonymized speakers in a scene using information from all previous scenes. While many studies have utilized TV show scripts to create such datasets, only `DialSim` includes unanswerable questions, conducts real-time evaluations, and fully utilizes the extended context of scripts to assess conversational agents.

## 3 DIALSIM

Our simulator, illustrated in Figure 1, features an agent taking on the role of a main character in a dialogue (*i.e.*, Friends: Ross, The Big Bang Theory: Sheldon, The Office: Michael[1]), encompassing around 350,000 tokens. Throughout the simulation, an agent is randomly asked questions by other characters that must be answered accurately within a time limit (§ 3.1). For each session, we prepare a large pool of questions in advance, from which one is randomly selected and presented to the agent every time a new simulation is initiated, simulating an environment for randomized tests (§ 3.2).

### 3.1 TASK

#### 3.1.1 DEFINITION

Let the $k$-th utterance of the $n$-th session be denoted as $u_{n,k}$, and the $n$-th session consisting of $r$ utterances be $\mathcal{S}_n = \{\{u_{n,1}, u_{n,2}, ..., u_{n,r}\}, d_n\}$, where $d_n$ is the date of $\mathcal{S}_n$. The sub-session including up to the $k$-th utterance of the $n$-th session is $\mathcal{S}_{n,k} = \{\{u_{n,1}, u_{n,2}, ..., u_{n,k}\}, d_n\}$. The entire dialogue consisting of $N$ sessions is denoted as $\mathcal{D} = \{\mathcal{S}_1, \mathcal{S}_2, ..., \mathcal{S}_N\}$. The agent's memory up to the $k$-th utterance of the $n$-th session is $\mathcal{M}_{n,k}$. The agent answering question $q_{n,m,c}$ asked by character $c$ in the $m$-th utterance of the $n$-th session using the memory is $a_{n,m} = Agent(\mathcal{M}_{n,m}, q_{n,m,c})$.

---

[1]The characters with the most lines in each script were selected.

---

**Algorithm 1** `DialSim`

---

**Input:** Dialogue $\mathcal{D} = \{\mathcal{S}_1, \mathcal{S}_2, \ldots, \mathcal{S}_N\}$, Time interval $t$, Agent
**Output:** CorrectAnswers / TotalQuestions
 1: CorrectAnswers $\leftarrow 0$
 2: TotalQuestions $\leftarrow 0$
 3: $\mathcal{M}_{1,0} \leftarrow \phi$
 4: **for** $n \leftarrow 1$ **to** $N$ **do**
 5:    **if** $|Characters(\mathcal{S}_n)| < 2$ **then**
 6:       **continue**
 7:    **else**
 8:       $u_{n,m} \leftarrow QuestionTimingSelection(\mathcal{S}_n)$
 9:       $c \leftarrow RandomCharacterInThreeTurns(u_{n,m})$
 10:      $q_{n,m,c}, TrueAnswer \leftarrow RandomQuestion(n, m, c)$
 11:      TotalQuestions $\leftarrow$ TotalQuestions + 1
 12:      **for** $k \leftarrow 1$ **to** $|\mathcal{S}_n|$ **do**
 13:        $\mathcal{M}_{n,k} \leftarrow UpdateMemoryInTime(\mathcal{M}_{n,k-1}, u_{n,k}, d_n, t)$
 14:        **if** $k = m$ **then**
 15:         $a_{n,m} \leftarrow AgentAnswerInTime(\mathcal{M}_{n,m}, q_{n,m,c}, d_n, t)$
 16:         **if** $a_{n,m} =$ TrueAnswer **then**
 17:           CorrectAnswers $\leftarrow$ CorrectAnswers + 1
 18:         **end if**
 19:        **end if**
 20:        $\mathcal{M}_{n+1,0} \leftarrow \mathcal{M}_{n,k}$
 21:      **end for**
 22:    **end if**
 23: **end for**

---

### 3.1.2 SIMULATOR

Algorithm 1 outlines the simulation process of `DialSim`, designed to emulate a real-time conversation. In this simulator, each participant's utterance (including the agent's) occurs at a predefined time interval (same as time limit), and the agent should update its memory[2] within this interval. If updating the memory is not completed within the interval, the simulator will move on to the next utterance (Line 13). During the simulation, other characters ask questions to the agent (Line 8-10), except in sessions where the agent is the only one talking (Line 5-6). The timing to ask a question is chosen randomly within the session (Line 8), and the speaker who asks the question is also chosen randomly. However, to make the simulation realistic, it is crucial to ensure that the chosen speaker is still present and hasn't left the session. We achieved this by randomly choosing from characters who were present within three turns of the agent's last utterance (Line 9). Then, a question is randomly selected and asked in the style of the corresponding speaker (Line 10). The agent then must respond to the question using its memory, all within the time limit (Line 15). The prompt for the response is created by combining the question with the dialogue history stored in the memory. If the response is not completed within the time limit, it will be considered a failure, and the simulator will move on to the next utterance. The prompt we used is provided in Appendix A.

### 3.2 DATA CONSTRUCTION

`DialSim` was developed using scripts from five consecutive seasons of popular TV shows (*i.e.*, Friends, The Big Bang Theory, and The Office[3]). These scripts were first preprocessed to serve as dialogue data (§ 3.2.1). Next, questions were generated for each script, drawing from fan quizzes (§ 3.2.2) and a temporal knowledge graph (TKG) (§ 3.2.3). Each question was then paired with

---

[2]The memory can be incrementally updated in various ways (*e.g.*, as per each utterance, a summary of each session up until the current utterance). A detailed discussion of these methods is provided in § 4.2.

[3]These shows were selected not only for having long stories spanning over five seasons but also to reflect a diverse range of real-world scenarios, including interpersonal relationships (Friends), academic conversations (The Big Bang Theory), and office-based interactions (The Office). The scripts were downloaded from the website Kaggle (https://www.kaggle.com/).

**(1) Web Crawling**

| Q# | Season | Episode | Question | Answer |
|----|--------|---------|----------|--------|
| 1 | 1 | 1 | What is the setting of the first scene? | Central Perk |
| 2 | 1 | 1 | How did Rachel buy her new boots in this episode? | Dad's credit card |

...

**(2-a) Filtering & Revision**
**(2-b) Evidence Scene Mapping**

| Q# | Season | Episode | Question | Evidence Scenes (2-b) |
|----|--------|---------|----------|------------------------|
| 1 | 1 | 1 | ~~What is the setting of the first scene?~~ **(2-a) Filtering** | REMOVED (visual content) |
| 2 | 1 | 1 | How did Rachel buy her new boots? ~~in this episode~~ **(2-a) Revision** | 14 (Date: September 22, 1994) |

...

**(3-a) Adding Date Info.**
**(3-b) Scene Annotation**

| Q# | Question | Answerable Scenes (3-b) | Unanswerable Scenes (3-b) |
|-----|----------|--------------------------|----------------------------|
| 2 | How did Rachel buy her new boots? | Scenes after 14 in Episode 1 | Scenes before 14 in Episode 1 |
| 2-1 | How did Rachel buy her new boots **on September 22, 1994**? **(3-a)** | Scenes after 14 | - |

...

**(4) Character Style Transfer**

| Q# | Character | Question |
|----|-----------|----------|
| 2 | Monica | Hey, do you know what Rachel used to snag those super cute new boots she's been rocking? |
| 2 | Joey | Hey, how did Rachel manage to snag those killer boots, huh? |

...

Figure 2: The overall process of question generation based on fan quizzes. First, we crawled fan quizzes from the web (1). Then, we applied filtering and revision processes to the crawled data (2-a, b). From this, we created secondary versions of the questions by adding dates to each (3-a). Then, we mapped each question to the scenes by determining whether it is answerable in that scene or not (3-b).[4] Finally, we applied character style transfer to make the questions more natural (4).

multiple choices and the correct answer. Finally, character style transfer was applied to refine the questions, resulting in the final pool of questions for each session (§ 3.2.4).

### 3.2.1 SCRIPT PREPROCESSING

The script we used includes 5 consecutive seasons per TV show, with each season containing approximately 20 episodes. Each episode is composed of multiple scenes (*i.e.*, session). Each script includes not only utterances but also descriptions of characters' actions and scenes, as well as metadata unrelated to the plot (*e.g.*, names of writers and directors). We manually filtered out all irrelevant parts to create $Script_{pre}$, which contains only the conversations between characters. Additionally, since some of our questions involve time conditions (*e.g.*, "Which friend wasn't allowed to drive Monica's Porsche in October 1994?"), we manually assigned a date to each scene in $Script_{pre}$ to provide time information to the agent. These dates were determined based on the contents of the conversations and the air dates of the episodes. The specific rules for date assignments are detailed in Appendix B. We then selected scenes involving the main character (*i.e.*, Ross, Sheldon, and Michael) from $Script_{pre}$ and sequentially numbered them as sessions $S_i$. This process resulted in the final dialogue $\mathcal{D} = \{S_1, S_2, ..., S_N\}$.

### 3.2.2 QUESTION GENERATION BASED ON FAN QUIZZES

We utilized a fan quiz website FunTrivia[5] to generate our questions. Fan quizzes cover a range of difficulty levels and focus on major events from each episode, making them promising for evaluating dialogue comprehension. Figure 2 illustrates our process for generating questions using fan quizzes. We began by extracting episode-specific quizzes from the site. Since these quizzes were created by dedicated fans, many required knowledge unrelated to the dialogue itself (*e.g.*, "What is the name of the actor who played the clerk?"). To filter out these questions, we first selected quizzes that could be answered by referencing $Script_{pre}$ using ChatGPT-4 (OpenAI, 2023b).[6] Additionally, ChatGPT-4 annotated the scenes that served as evidence for each question. These annotations were verified by the authors to ensure accurate filtering and scene-mapping.

---

[4]Questions without dates (*e.g.*, Q#2 in Figure 2) are episode-specific, so the answers to such questions can vary across episodes. Therefore, for questions without dates, we mapped them only to the scenes within the episode to which the question belongs. On the other hand, questions with specified dates (*e.g.*, Q#2-1 in Figure 2) are episode-agnostic. However, asking them on dates before the specified dates in the question would not be natural; thus we did not map such scenes to such questions.

[5]https://www.funtrivia.com/

[6]Fan quizzes exist for each episode, so we annotated them based on $Script_{pre}$. Since $Script_{pre}$ contains scenes where the main character is absent, questions about these scenes would be unanswerable, which allowed us to design adversarial tests. Therefore, we first generated questions for each scene based on $Script_{pre}$ and then matched them to the sessions of $\mathcal{D}$.

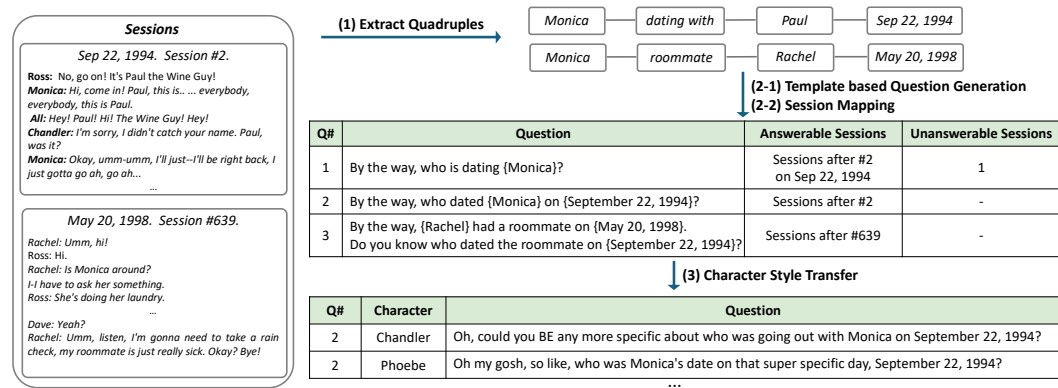

Figure 3: The overall process of question generation based on the temporal knowledge graph. We first extracted quadruples and constructed a temporal knowledge graph (1). Then, we generated questions based on this and mapped each question to the sessions by determining whether it was answerable in that session or not, similar to fan quiz-based questions (2-1, 2-2). Character style transfer was performed afterwards (3).

We then annotated the answerability of each question, i.e., whether it is possible for the main character to know the answer in the corresponding scene. For example, in Friends, if the evidence for a question was in scene 14, Ross would not know the answer if he was absent from that scene. Even if he were present in scene 14, he couldn't answer the question if it was asked in scene 1. However, if Ross appeared in scene 14 and the question was then asked in scene 15, he would know the answer. Using this principle, we determined whether each question is answerable. Additionally, to create questions that require long-term memory, new questions were generated by adding the date information of each scene to the questions (*e.g.*, "How did Rachel buy her new boots on September 22, 1994?"). Detailed question generation processes are provided in Appendix C.

### 3.2.3 QUESTION GENERATION BASED ON A TEMPORAL KNOWLEDGE GRAPH

Fan quizzes are useful for generating our questions, but since they are episode-specific and user-generated, the questions don't span multiple episodes and their numbers are limited (∼1K). To address this, we constructed a knowledge graph for each session and used it to generate questions. Initially, we used ChatGPT-4 to extract triples (*i.e.*, [head, relation, tail]) from each session $S_i$ in $\mathcal{D}$. These triples were then refined by the authors. We employed 32 relations (*e.g.*, girlfriend) derived from DialogRE (Yu et al., 2020), a high-quality dataset where human annotators manually extracted relations from Friends scripts, classifying relationships between characters into 37 categories. We adapted and modified these relations for our purpose. More details about the relations are provided in Appendix D.1. Finally, we combined the triples from each session with their respective dates to create a temporal knowledge graph (TKG) composed of quadruples (*i.e.*, [head, relation, tail, date]).

Using the constructed TKG, we created questions that the main character could either answer or not for each session. We generated these questions by extracting one (*i.e.*, one-hop) or two (*i.e.*, two-hop) quadruples from the TKG. The form and answer of the question may change depending on the time it is asked, even if the same quadruple is used. For instance, if we select [Rachel, boyfriend, Ross, 1994-08-08] and ask the question in 1996, it would be: "Who was Rachel's boyfriend on August 8th, 1994?" If asked on August 8th, 1994, the question would be: "Who is Rachel's boyfriend?" In both cases, the answer is Ross. Conversely, if we inquire about Rachel's boyfriend in 1992, when no information is available, the correct answer would be: "I don't know." In this manner, we manually verified the answer of each question. We applied the same principle to create more complex two-hop questions (*e.g.*, "Rachel had a roommate on August 8th, 1994. Who is the boyfriend of the roommate now?"). The overall process of generating questions using TKG is illustrated in Figure 3. Examples of question templates and corresponding questions we created can be found in Appendix D.2.

### 3.2.4 FINAL DATA PROCESSING

**Answer Choices Generation** To create multiple-choice questions, we carefully crafted a set of answer choices for each question. First, for all questions, we included a choice "(E) I don't know.",

Table 2: Statistics of `DialSim`. The values in parentheses for Average Question Candidates per Session refer to the number of answerable questions and unanswerable questions, respectively.

| | Friends | The Big Bang Theory | The Office |
|---|---|---|---|
| Total Number of Tokens | 335439 | 367636 | 352914 |
| Total Number of Sessions | 785 | 805 | 2338 |
| Average Fan Quiz Questions per Session | 56.7 | 7.8 | 12.7 |
| Average TKG Questions per Session | 1127.7 | 1158.5 | 508.5 |
| Average Question Candidates per Session | 1196.8 (1115.1 / 81.7) | 1196.9 (1011.0 / 185.9) | 511.5 (486.5 / 25.0) |
| Approximate Number of Possible Tests | $1196.8^{785}$ | $1196.9^{805}$ | $511.5^{2338}$ |

which agents must choose if the questions are unanswerable. For questions sourced from the fan quizzes, the four answer choices were taken from the original quiz. The correct answers for these questions were the same as the original quiz, while the unanswerable questions were fixed to (E). For questions based on the TKG, the incorrect choices were derived from the tails of other quadruples that shared the same relation as the original quadruple. For example, for the question "Who is Rachel's boyfriend?", we extracted quadruples from the whole TKG where the relation is "boyfriend" and randomly selected three tails to form the incorrect choices. Additionally, to create a more adversarial test, if Rachel has a boyfriend in the past or future, we prioritized including these in the incorrect choices. In this case, for answerable questions (*i.e.*, past or present), the correct answer is the tail of the original quadruple, while for unanswerable questions (*i.e.*, future), the correct answer is (E).

**Question Style Transfer**   In the simulator, questions are rephrased to reflect each character's unique tone, creating the impression that the characters themselves are asking the questions (*e.g.*, Generic style: "How did Rachel buy her new boots?"→ Style of Joey Tribbiani from Friends: "Hey, how did Rachel manage to snag those killer boots, huh?"). This transformation is powered by ChatGPT-4, and subsamples are reviewed by the authors to ensure that the original intent was preserved. More examples of style-transferred questions for each character can be found in Appendix E.

## 4 EXPERIMENTS

### 4.1 EXPERIMENTAL SETTING

To efficiently and accurately evaluate the agents' long-term dialogue understanding abilities, we used a multiple-choice format for the questions. Table 2 shows the statistics for `DialSim`, revealing a notable difference between the number of answerable and unanswerable questions. To ensure a balanced distribution of correct answers during the simulation, 20% of the questions were intentionally designed to be unanswerable, with each question offering five possible choices. In addition to the multiple-choice format, we also offer an option to use an open-ended format, allowing users to choose their preferred question format.

`DialSim` operates in real-time, requiring precise control of the experimental environment. Therefore, we conducted all experiments using the same hardware: NVIDIA RTX A6000 GPUs and an AMD EPYC 7702 64-Core Processor. The time limit used in the experiment was set to 6 seconds, based on the average time interval between utterances in the TV shows. Note that the time limit can be set to any value (even infinity) that meets one's service requirement. We provide extensive discussions on the time limit feature of `DialSim`, including the test environment control and internet speed in Appendix F, along with details about question formats.

### 4.2 BASELINES

We experimented with two methods for using an agent's memory. The first method, namely Base LLM, is to simply prefix latest utterances as much allowed by the model's context length. The second method, namely RAG-based, employs a retriever to search for relevant dialogue history from the agent's memory (external storage) and includes it in the prompt (Lewis et al., 2020). This method can be broken down into three ways for storing dialogue history: each speaker's utterance individually, the entire session, and a summarized version of each session (denoted as *Utterance*, *Session Entire*, and *Session Sum.* in Tables 3 and 4). The retrieval from the memory was performed using BM25 (Robertson et al., 2009) and cosine similarity with the OpenAI embeddings (OpenAI, 2024c). Agents were tested with both API-based models (*i.e.*, Gemini-1.0 Pro, 1.5 Pro (Team et al., 2023; Reid et al., 2024), Claude 3 Opus (Anthropic, 2024), ChatGPT-3.5, 4o, 4o-mini (OpenAI,

Table 3: The performance of the agents on Friends dialogue in `DialSim` (time limit = 6 seconds). We conducted experiments three times and reported the accuracy and standard deviations.

| Type | Model | Base LLM | RAG-based | | | | | |
|---|---|---|---|---|---|---|---|---|
| | | | BM25 | | | OpenAI Embedding | | |
| | | | Utterance | Session Entire | Session Sum. | Utterance | Session Entire | Session Sum. |
| API | ChatGPT-4o-mini | 38.53 (0.89)† | 32.65 (2.65) | 49.04 (1.67) | 40.27 (1.36) | 40.10 (0.75) | 44.36 (2.36) | 42.53 (1.26) |
| | ChatGPT-3.5 | 31.82 (1.31) | 25.58 (1.78) | 39.70 (1.86) | 32.09 (0.84) | 32.06 (1.60) | 36.84 (1.77) | 36.69 (1.25) |
| | Gemini 1.0 pro | 2.96 (0.31) | 28.77 (1.83) | 25.07 (2.40) | 35.27 (1.80) | 34.22 (0.49) | 31.83 (0.41) | 35.75 (2.93) |
| Open | Tulu2-70B | 0.37 (0.15) | 20.94 (0.75) | 20.27 (0.99) | 19.75 (0.08) | 31.76 (1.84) | 10.15 (0.55) | 18.87 (0.30) |
| | Tulu2-7B | 0.84 (0.15) | 12.68 (0.24) | 19.58 (1.04) | 26.84 (0.85) | 14.08 (0.89) | 17.39 (1.37) | 25.21 (1.28) |
| | Llama3.1-70B | 0.60 (0.06)† | 31.08 (1.21) | 0.55 (0.12) | 16.26 (5.05) | 39.00 (0.30) | 2.26 (0.42) | 20.14 (0.22) |
| | Llama3.1-8B | 28.82 (1.94)† | 27.12 (0.95) | 34.14 (0.85) | 30.91 (0.63) | 29.76 (1.31) | 33.25 (0.57) | 24.48 (0.60) |
| | Mixtral-8x7B | 1.88 (0.26) | 16.84 (0.95) | 26.23 (0.90) | 17.11 (1.94) | 17.94 (1.32) | 26.78 (1.04) | 15.40 (1.39) |
| | Mistral-7B | 2.82 (0.46) | 24.22 (2.04) | 33.07 (1.01) | 29.29 (1.76) | 28.30 (1.93) | 29.15 (1.67) | 25.41 (1.53) |
| | Gemma-7B | 16.60 (0.84) | 22.11 (1.73) | 24.30 (2.04) | 18.33 (1.37) | 26.42 (2.48) | 22.54 (0.78) | 18.80 (0.64) |
| | Gemma-2B | 0.68 (0.20) | 24.06 (2.03) | 24.22 (1.34) | 25.79 (1.00) | 25.31 (1.55) | 24.48 (1.62) | 25.78 (1.12) |

†: Both ChatGPT-4o-mini and Llama3.1 support up to 128k tokens, but we limited them to 8k tokens due to high costs and GPU VRAM limits, respectively.

Table 4: The performance of the agents on Friends dialogue in `DialSim` (without time limit). We conducted experiments three times and reported the accuracy and standard deviations.

| Type | Model | Base LLM | RAG-based | | | | | |
|---|---|---|---|---|---|---|---|---|
| | | | BM25 | | | OpenAI Embedding | | |
| | | | Utterance | Session Entire | Session Sum. | Utterance | Session Entire | Session Sum. |
| API | ChatGPT-4o-mini | 38.91 (0.99)† | 34.44 (0.52) | 49.21 (0.12) | 42.23 (1.57) | 38.91 (0.74) | 43.64 (0.42) | 42.40 (0.99) |
| | ChatGPT-3.5 | 31.81 (1.33) | 26.91 (2.30) | 39.45 (1.40) | 32.77 (1.31) | 32.41 (0.96) | 35.78 (0.74) | 35.98 (1.75) |
| | Gemini 1.0 pro | 28.36 (0.97) | 28.10 (1.08) | 39.90 (1.08) | 34.11 (1.64) | 34.26 (2.91) | 30.93 (2.17) | 33.96 (2.11) |
| Open | Tulu2-70B | 3.31 (0.32) | 29.87 (0.65) | 35.87 (2.56) | 34.72 (1.63) | 37.07 (0.72) | 33.63 (1.32) | 38.62 (1.94) |
| | Tulu2-7B | 1.57 (0.12) | 28.93 (2.81) | 28.72 (1.80) | 30.86 (2.29) | 34.55 (0.47) | 31.04 (0.96) | 32.12 (0.75) |
| | Llama3.1-70B | 36.36 (0.68)† | 31.84 (1.29) | 43.17 (0.99) | 43.81 (0.94) | 39.85 (2.08) | 43.17 (0.68) | 48.49 (0.97) |
| | Llama3.1-8B | 28.78 (0.34)† | 29.89 (1.56) | 34.70 (1.75) | 33.93 (1.76) | 31.63 (2.17) | 32.91 (0.51) | 35.59 (1.09) |
| | Mixtral-8x7B | 42.19 (1.76) | 31.84 (0.78) | 46.47 (1.75) | 32.31 (1.09) | 35.51 (0.19) | 41.24 (2.90) | 34.18 (0.96) |
| | Mistral-7B | 32.93 (0.59) | 28.20 (1.17) | 35.09 (1.76) | 30.16 (1.82) | 30.12 (1.45) | 31.00 (1.93) | 30.80 (1.75) |
| | Gemma-7B | 18.78 (0.87) | 22.26 (1.52) | 23.62 (2.09) | 19.83 (1.74) | 25.07 (0.49) | 22.48 (0.25) | 20.08 (0.76) |
| | Gemma-2B | 1.16 (0.26) | 25.03 (1.54) | 24.64 (1.31) | 24.84 (2.05) | 28.06 (1.38) | 24.56 (2.60) | 28.28 (1.94) |

†: Both ChatGPT-4o-mini and Llama3.1 support up to 128k tokens, but we limited them to 8k tokens due to high costs and GPU VRAM limits, respectively.

2023a; 2024a;b))[7] and open-source models (*i.e.*, Tulu2-7B, 70B (Ivison et al., 2023), Llama3.1-8B, 70B (Meta, 2024), Mistral-7B, 8x7B (Jiang et al., 2023; 2024), and Gemma-2B, 7B (Team et al., 2024)). To emulate conversational settings, we used chat templates for instruction-tuned models or directly used chat models.

## 4.3 RESULTS

Table 3 shows that API-based models outperformed open-source models due to their superior inference capabilities and faster response times. However, the performances of all baselines were below 50%, suggesting that current LLMs function poorly as conversational agents for multi-party long-term complex dialogues. The experimental results for Friends, The Big Bang Theory, and The Office exhibited similar trends. The detailed results are described in Appendix G.

**Time Limit** We conducted additional experiments without time limit and reported the results in Table 4. Under time limit, there were often no significant differences in performances based on model size when comparing the same type of LLM, and sometimes smaller models outperformed larger ones. However, in the absence of time limit, larger models typically exhibited better performances than their smaller counterparts. Additionally, larger open-source models have shown remarkable inference capabilities, often on par with the performance of API-based models. This is because larger models often exceed the time limit due to their slower inference speed. Therefore, for a conversational agent to engage effectively in real-time conversations, it is crucial to select a model size that balances both inference time and reasoning capability. The performances according to different time limits are reported in Appendix H.

---

[7]Gemini-1.5 Pro, Claude 3 Opus, and ChatGPT-4o were evaluated only in the BM25-Session Entire and oracle setting to measure their performance upper bound due to their high prices. The experimental results can be found in Appendix I.

**Storing History**  Storing the entire session consistently exhibits superior performance compared to other history storing methods, because individual utterances lack adequate context, and crucial information may be lost during summarization. However, Llama3.1-70B achieved the best performance when using Session Sum. as a history saving method, owing to its strong summarization capabilities. Additionally, contrary to our expectations, Mixtral's Base LLM (*i.e.*, without history retrieval) outperforms some retrieval-based models in settings with unlimited time. This is due to Mixtral's context length of 32k tokens, which is long enough to accommodate half a season of the script, allowing it to utilize a longer dialogue history than some of the other baselines. However, in a setting with a time limit, Mixtral's performance significantly drops due to its long inference time. Therefore, for a conversational agent to converse in real-time, it is necessary to select a reasonably appropriate length of dialogue history.

**Oracle Setting**  To establish a performance upper bound, we conducted experiments in an oracle setting, where the agents were provided with the evidence sessions and their dates (see Figure 2). In this scenario, Llama3.1-70B achieved the best performance of 69.86% in a setting with unlimited time, highlighting the challenging nature of our task. This result surpasses the best RAG-based method, OpenAI Embedding-Session Sum. setting, by 21.37%. This notable gap underscores the necessity for advanced techniques in storing and retrieving history for agents engaged in long-term dialogues. Detailed experimental results are provided in Appendix I.

**Error Analysis by Question Type**  We conducted an error analysis by question type on ChatGPT-4o-mini, based on BM25-Session Entire, which showed the highest performance setting without a time limit. First, comparing the performance of fan quiz-based questions and TKG-based questions, the results were 58.80% and 46.42% respectively, indicating the greater difficulty of TKG-based questions. Additionally, within TKG-based questions, one-hop questions had a performance of 66.67%, whereas two-hop questions had a performance of 13.53%, highlighting the challenge of two-hop questions. Furthermore, even in the oracle setting, while the performance of one-hop questions increased to 84.05%, two-hop questions remained at 28.45%. This suggests that two-hop questions are challenging not only in terms of history retrieval but also in reasoning across the given sessions.

**Adversarial Test**  LLM-based agents are likely to have prior knowledge about these TV shows from the pre-training process (see Appendix K). Since such agents can provide answers without referring to the actual dialogue history, it is crucial to ensure that the agent relies strictly on the history for its responses. To achieve this, we conducted further experiments for the adversarial test by altering the names of the characters in two ways: by swapping their names with each other (*e.g.*, Joey ↔ Monica) or by assigning new names to them (*e.g.*, Joey → John). The results showed a noticeable decrease in overall performance compared to the original setup. Specifically, when we experimented under unlimited time conditions, performance dropped by 5% to 10%. This decline is attributed to the agents relying not only on the dialogue history but also on their pre-trained knowledge when answering questions. Additionally, the performance decrease was more pronounced when names were swapped compared to when new names were assigned. This suggests that new names represent new information, while mixed names in the dialogue history conflicted with the pre-trained knowledge, leading to reduced reasoning ability. The detailed experimental results are provided in Appendix J.

## 5  CONCLUSION

In this paper, we introduce `DialSim`, a simulator for evaluating conversational agents' long-term dialogue understanding in real-time settings. `DialSim` utilizes scripts from well-known TV shows and incorporates questions derived from fan quizzes, along with a temporal knowledge graph, for a thorough assessment. Our experimental findings reveal significant limitations in current conversational agents' abilities to manage complex, multi-party, long-term dialogues effectively.

Despite its strengths, our simulator has two main limitations. First, while the questions and answers are logically paired for accurate evaluation, the random selection of questions could introduce a bit of awkwardness during conversations. Second, while we considered incorporating industry-specific dialogues such as chat logs from customer service or retail, where conversational agents could be used for business purposes, these dialogue datasets are usually proprietary and not publicly accessible. In future developments, we will focus on enhancing the natural flow of interactions and creating simulators that are applicable to real-world industries.

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

## A    PROMPT FOR RESPONSE GENERATION

The prompt given to the conversational agent to answer questions using dialogue history is shown in Table 5. An example where the placeholders from Table 5 are filled with actual values can be found in Table 6.

Table 5: In the $<<<$Chatbot$>>>$ placeholder, the name of the main character (*i.e.*, Ross, Sheldon, Michael) for each TV show is inserted. In the $<<<$Date$>>>$ placeholder, the date of the session in which the question is being asked is inserted. In the $<<<$Dialog_History$>>>$ placeholder, the dialogue history that the agent will use is inserted. In the $<<<$Question$>>>$ placeholder, the question that the agent should answer along with five choices is inserted.

| Prompt for Response Generation |
| --- |
| You are $<<<$Chatbot$>>>$, a long-term conversational agent capable of interacting with multiple users. Based on the [Retrieved Dialogue History] provided, please answer the given [Question]. Note the following points: 1. Your answer must exclusively be one of the options: (A), (B), (C), (D), (E). 2. Your responses should solely rely on the retrieved dialogue history. If the information in the dialogue history is insufficient to answer the question, you must choose (E). 3. This question is being asked in the context of $<<<$Date$>>>$. [Retrieved Dialogue History] $<<<$Dialog_History$>>>$ [Question] $<<<$Question$>>>$ [Answer] |

## B    DATE ASSIGNMENT

We first extracted elements from the scripts that could indicate dates (*e.g.*, Valentine's Day, Christmas Eve). Then, we reviewed the scripts again to analyze the relative timing of the sessions. For example, if there is a line mentioning that Chandler broke up with his girlfriend two days ago, we annotated the session where he broke up with his girlfriend as occurring two days prior to the mentioned session. Next, while watching each episode, we pinpointed sessions where the dates might have changed by observing whether the characters' outfits changed between sessions. Finally, we assigned a specific date to each session based on the actual broadcast date of the episode, adjusting for the relative differences in dates and events such as Christmas.

## C    QUESTION GENERATION BASED ON FAN QUIZZES

For each scene $s_{i,k}$ from episode $p_i$ in $Script_{pre}$, we define the set of answerable questions as $FanA_{i,k}$ and the set of unanswerable questions as $FanU_{i,k}$. The process of generating questions based on fan quizzes is as follows.

First, we collected quizzes for each season and episode of Friends, The Big Bang Theory, and The Office from the FunTrivia website. For each episode $p_i$ in $Script_{pre}$, we used ChatGPT-4 to determine if the crawled questions $CrQ_i = \{q_{i,0}, q_{i,1}, ..., q_{i,l}\}$ could be answered using only $p_i$. If a question $q_{i,m}$ could be answered, ChatGPT-4 identified the scenes $ES_{i,m}$ that provide evidence for the answer, compiling them into $Q_i = \{(q_{i,m}, ES_{i,m})\}_{m=0}^{l}$. Subsequently, the authors reviewed each $ES_{i,m}$, made necessary corrections, and annotated whether a single scene from $ES_{i,m}$ was sufficient to answer $q_{i,m}$ or if multiple scenes were needed to be considered simultaneously. For each $s_{i,k}$ within $p_i$, we assessed the answerability of the questions in $Q_i$.

For each $s_{i,k}$, if a question $q_{i,m}$ could be answered using just one scene, and $s_{i,k}$ occurs after the initial appearance of the main character in $ES_{i,m}$, we included $q_{i,m}$ in $FanA_{i,k}$. This ensures that the main character had adequate exposure to the relevant evidence. Additionally, for questions requiring verification across multiple scenes, if the main character appears in all $ES_{i,m}$ scenes and $s_{i,k}$ occurs

Table 6: An actual example of the prompt for response generation.

| Prompt for Response Generation |
|---|
| You are Ross, a long-term conversational agent capable of interacting with multiple users. |
| Based on the [Retrieved Dialogue History] provided, please answer the given [Question]. |
| Note the following points: |
| 1. Your answer must exclusively be one of the options: (A), (B), (C), (D), (E). |
| 2. Your responses should solely rely on the retrieved dialogue history. If the information in the dialogue |
| history is insufficient to answer the question, you must choose (E). |
| 3. This question is being asked in the context of [February 26, 1999]. |
| |
| [Retrieved Dialogue History] |
| |
| [Session #1 on September 22, 1994] |
| <<Session Omitted>> |
| Ross: No, go on! It's Paul the Wine Guy! |
| Phoebe: What does that mean? Does he sell it, drink it, or just complain a lot? |
| Monica: Hi, come in! Paul, this is.. ... everybody, everybody, this is Paul. |
| All: Hey! Paul! Hi! The Wine Guy! Hey! |
| Chandler: I'm sorry, I didn't catch your name. Paul, was it? |
| Monica: Okay, umm-umm, I'll just–I'll be right back, I just gotta go ah, go ah. . . |
| Ross: A wandering? |
| Monica: Change! Okay, sit down. Two seconds. |
| Phoebe: Ooh, I just pulled out four eyelashes. That can't be good. |
| <<Session Omitted>> |
| |
| [Session #2 on May 20, 1998] |
| <<Session Omitted>> |
| Rachel: Umm, hi! |
| Ross: Hi. |
| Rachel: Is Monica around? I-I have to ask her something. |
| Ross: She's doing her laundry. |
| <<Session Omitted>> |
| Rachel: Y'know what Ross? You're not going anywhere. You're gonna sit right here. |
| I'm gonna make you a cup of tea and we're gonna talk this thing whole out. All right? Hey, Dave! |
| Dave: Yeah? |
| Rachel: Umm, listen, I'm gonna need to take a rain check, my roommate is just really sick. |
| Okay? Bye! Honey, listen, I know, I know things seem so bad right now. |
| |
| [Question] Chandler: So, just for a little stroll down memory lane, Rachel was bunking with someone in May 1998. |
| Any wild guesses on who was dating this mystery cohabitant by September 22, 1994? |
| (A) Paolo (B) Paul (C) Roger (D) Vince (E) I don't know. |
| [Answer] |

after the last scene of $ES_{i,m}$, we included $q_{i,m}$ in $FanA_{i,k}$. If the main character does not appear in any of the $ES_{i,m}$ scenes, $q_{i,m}$ was included in $FanU_{i,k}$ since the main character has not experienced any evidence to answer the question. The rest are not included in the dataset as it is unclear whether they are answerable per scene. Additionally, to generate questions that require long-term memory, we added the most recent date of the evidence scenes for each question.

# D QUESTION GENERATION BASED ON A TEMPORAL KNOWLEDGE GRAPH

## D.1 RELATIONS

We used the following 32 relations: 'age', 'alumni', 'boss', 'boyfriend', 'brother', 'client', 'date of birth', 'dating with', 'ex-boyfriend', 'ex-fiance', 'ex-fiancee', 'ex-girlfriend', 'ex-husband', 'ex-roommate', 'ex-wife', 'father', 'fiance', 'fiancee', 'girlfriend', 'hometown', 'husband', 'job', 'major',

'mother', 'neighbor', 'pet', 'place of birth', 'place of work', 'roommate', 'sister', 'subordinate', 'wife'.

## D.2    QUESTION TEMPLATES AND GENERATED QUESTIONS

Templates for one-hop questions are provided in Table 7 and Table 8. The former contains templates without temporal information, while the latter includes templates with temporal details. Since relations like "brother" and "sister" remain constant over time, questions about these relations do not require temporal information. Hence, no temporal templates were created for them. In Table 8, "on {time}" is used, but {time} can be not only the full date (year, month, and day) but also just the year and month, or even just the year. In these cases, "in {time}" is used.

The templates for two-hop questions are available in Table 9. These templates incorporate temporal information. To frame questions in the present tense, adjust the verbs to the present tense and remove the temporal information, following the approaches demonstrated in Table 7 and Table 8.

Table 7: Templates for one-hop questions without temporal information.

| Question Type | Relation | Template | Question Example |
|---|---|---|---|
| | alumni | Who is {sub}'s alumni? | Who is Lincoln High School's alumni? |
| | boss | Who is {sub}'s boss? | Who is Chandler's boss? |
| | subordinate | Who is {sub}'s subordinate? | Who is Chandler's subordinate? |
| | client | Who is {sub}'s client? | Who is Chandler's client? |
| | neighbor | Who is {sub}'s neighbor? | Who is Chandler's neighbor? |
| | roommate | Who is {sub}'s roommate? | Who is Chandler's roommate? |
| | ex-roommate | Who is {sub}'s ex-roommate? | Who is Chandler's ex-roommate? |
| | fiance | Who is {sub}'s fiance? | Who is Rachel's fiance? |
| | fiancee | Who is {sub}'s fiancee? | Who is Ross's fiancee? |
| | ex-fiance | Who is {sub}'s ex-fiance? | Who is Rachel's ex-fiance? |
| | ex-fiancee | Who is {sub}'s ex-fiancee? | Who is Ross's ex-fiancee? |
| | pet | Who is {sub}'s pet? | Who is Ross's pet? |
| | dating with | Who is dating {sub}? | Who is dating Ross? |
| | job | What is {sub}'s job? | What is Ross's job? |
| | place of work | Where does {sub} work? | Where does Ross work? |
| **Without Time** | age | How old is {sub}? | How old is Ross? |
| | major | What is {sub}'s major? | What is Ross's major? |
| | mother | Who is {sub}'s mother? | Who is Ross's mother? |
| | father | Who is {sub}'s father? | Who is Ross's father? |
| | place of birth | Where was {sub} born? | Where was Ben born? |
| | hometown | Where is {sub}'s hometown? | Where is Monica's hometown? |
| | date of birth | When was {sub} born? | When was Ben born? |
| | husband | Who is {sub}'s husband? | Who is Emily's husband? |
| | wife | Who is {sub}'s wife? | Who is Ross's wife? |
| | girlfriend | Who is {sub}'s girlfriend? | Who is Joey's girlfriend? |
| | boyfriend | Who is {sub}'s boyfriend? | Who is Monica's boyfriend? |
| | ex-husband | Who is {sub}'s ex-husband? | Who is Carol's ex-husband? |
| | ex-wife | Who is {sub}'s ex-wife? | Who is Ross's ex-wife? |
| | ex-girlfriend | Who is {sub}'s ex-girlfriend? | Who is Ross's ex-girlfriend? |
| | ex-boyfriend | Who is {sub}'s ex-boyfriend? | Who is Rachel's ex-boyfriend? |
| | brother | Who is {sub}'s brother? | Who is Monica's brother? |
| | sister | Who is {sub}'s sister? | Who is Ross's sister? |

Table 8: Templates for one-hop questions with temporal information.

| Question Type | Relation | Template | Question Example |
|---|---|---|---|
| | boss | Who was {sub}'s boss on {time}? | Who was Chandler's boss on September 26th, 1994? |
| | client | Who was {sub}'s client on {time}? | Who was Chandler's client on September 26th, 1994? |
| | neighbor | Who was {sub}'s neighbor on {time}? | Who was Chandler's neighbor on September 26th, 1994? |
| | roommate | Who was {sub}'s roommate on {time}? | Who was Chandler's roommate on September 26th, 1994? |
| | fiance | Who was {sub}'s fiance on {time}? | Who was Rachel's fiance on September 26th, 1994? |
| | fiancee | Who was {sub}'s fiancee on {time}? | Who was Ross's fiancee on September 26th, 1994? |
| | pet | Who was {sub}'s pet on {time}? | Who was Ross's pet on September 26th, 1994? |
| **With Time** | dating with | Who dated {sub} on {time}? | Who dated Ross on September 26th, 1994? |
| | job | What was {sub}'s job on {time}? | What was Monica's job on September 26th, 1994? |
| | place of work | Where did {sub} work on {time}? | Where did Monica work on September 26th, 1994? |
| | age | How old was {sub} on {time}? | How old was Monica on September 26th, 1994? |
| | major | What was {sub}'s major on {time}? | What was Ross's major on September 26th, 1994? |
| | husband | Who was {sub}'s husband on {time}? | Who was Emily's husband on September 26th, 1994? |
| | wife | Who was {sub}'s wife on {time}? | Who was Ross's wife on September 26th, 1994? |
| | girlfriend | Who was {sub}'s girlfriend on {time}? | Who was Ross's girlfriend on September 26th, 1994? |
| | boyfriend | Who was {sub}'s boyfriend on {time}? | Who was Rachel's boyfriend on September 26th, 1994? |

Table 9: Templates for two-hop questions.

| First Relation | Second Relation | Template | Question Example |
|---|---|---|---|
| roommate, wife, husband, girlfriend, boyfriend, client, neighbor, boss, subordinate, fiance, fiancee | roommate, wife, husband, pet, girlfriend, boyfriend, client, neighbor, boss, subordinate, fiance, fiancee | {sub1} had a {First Relation} on {time1}. Who was the {Second Relation} of the {First Relation} on {time2}? | Monica had a roommate on September 26th, 1994. Who was the boyfriend of the roommate on October 5th, 1996? |
| | dating with | {sub1} had a {First Relation} on {time1}. Who dated the {First Relation} on {time2}? | Monica had a roommate on September 26th, 1994. Who dated the roommate on October 5th, 1996? |
| | job, major, age | {sub1} had a {First Relation} on {time1}. What was the {Second Relation} of the {First Relation} on {time2}? | Monica had a roommate on September 26th, 1994. What was the job of the roommate on October 5th, 1996? |
| | mother, father, son, daughter, sister, brother | {sub1} had a {First Relation} on {time1}. Who is the {Second Relation} of the {First Relation}? | Monica had a roommate on September 26th, 1994. Who is the mother of the roommate? |
| | date of birth, place of birth, | {sub1} had a {First Relation} on {time1}. When (Where) was the {First Relation} born? | Monica had a roommate on September 26th 1994. When was the roommate born? |
| | place of work | {sub1} had a {First Relation} on {time1}. Where did the {First Relation} work on {time2}? | Monica had a roommate on September 26th, 1994. Where did the roommate work on October 5th, 1996? |
| | hometown | {sub1} had a {First Relation} on {time1}. Where is the hometown of the {First Relation}? | Monica had a roommate on September 26th, 1994. Where is the hometown of the roommate? |
| dating with | roommate, wife, husband, girlfriend, boyfriend, client, neighbor, boss, subordinate, fiance, fiancee | {sub1} dated a person on {time1}. Who was the {Second Relation} of the person on {time2}? | Monica dated a person on September 26th, 1994. Who was the boss of the person on October 5th, 1996? |
| mother, father, son, daughter, sister, brother | roommate, wife, husband, girlfriend, boyfriend, client, neighbor, boss, subordinate, fiance, fiancee | Who was the {Second Relation} of {sub1}'s {First Relation} on {time2}? | Who was the roommate of Ross's sister on September 26th, 1994? |
| | dating with | Who dated {sub1}'s {First Relation} on {time2}? | Who dated Ben's father on September 26th, 1994? |
| | job, age, major | What was the {Second Relation} of {sub1}'s {First Relation} on {time2}? | What was the job of Ben's father on September 26th, 1994? |
| | mother, father, son, daughter, sister, brother | Who is the {Second Relation} of {sub1}'s {First Relation}? | Who is the mother of Ross's son? |
| | date of birth, place of birth | When (Where) was {sub1}'s {First Relation} born? | When was Monica's brother born? |
| | place of work | Where did {sub1}'s {First Relation} work on {time2}? | Where did Monica's brother work on October 5th, 1996? |
| | hometown | Where is the hometown of {sub1}'s {First Relation}? | Where is the hometown of Ross's son? |

# E    CHARACTER STYLE TRANSFER

Table 10 shows the results of the character style transfer for three selected questions. To make the questions sound more natural and conversational, we prepended each one with "By the way,". This helps them blend seamlessly into the flow of the conversation. The table shows how each question appears when rephrased in the style of various characters. The 'Default' setting is applied when the question is asked by a character who is not a recurring character of the TV show.

Table 10: Examples of the results of character style transfer.

| Original Question | Character | Style Transferred Question |
|---|---|---|
| By the way, how did Rachel buy her new boots? | Default | Hey, any idea what Rachel used to snag those stylish new boots of hers? |
| | Monica | Hey, do you know what Rachel used to snag those super cute new boots she's been rocking? |
| | Chandler | So, could we BE any more curious about how Rachel snagged those new boots? |
| | Joey | Hey, how did Rachel manage to snag those killer boots, huh? |
| | Phoebe | Oh my gosh! Do you have any idea how Rachel snagged those super cute new boots? |
| By the way, who dated Monica on September 22, 1994? | Default | So, who was Monica's date on the night of September 22, 1994? |
| | Chandler | Oh, could you BE any more specific about who was going out with Monica on September 22, 1994? |
| | Joey | Hey, just outta curiosity, who was goin' out with Monica on September 22, 1994? |
| | Phoebe | Oh my gosh, so like, who was Monica's date on that super specific day, September 22, 1994? |
| | Rachel | Oh my god, so like, who was going out with Monica on September 22, 1994?' |
| By the way, Rachel had a roommate on October 28, 1994. Who dated the roommate in September 1994? | Default | Oh. My. God. Remember when Rachel had a roommate back on October 28, 1994? So, who was going out with that roommate by September 1994? |
| | Monica | Hey, just out of curiosity, do you know who was going out with Rachel's roommate from back in September 1994? I remember she got that roommate around October 28, 1994. |
| | Chandler | So, just for a little stroll down memory lane, Rachel was bunking with someone on October 28, 1994. Any wild guesses on who was dating this mystery co-habitant by September 1994? |
| | Joey | Hey, so you know how Rachel was living with someone back on October 28, 1994, right? So I'm just wonderin' here, who was going out with this roommate of hers in September 1994? |
| | Phoebe | By the way, Rachel had a roommate on October 28, 1994. Who dated the roommate in September 1994? |

# F EXPERIMENTAL SETTING

## F.1 TIME LIMIT

In `DialSim`, the time limit is a controllable parameter, giving developers the flexibility to conduct experiments with any chosen time constraint, or even without one. When a time limit is set, the experimental environment can impact performance. Consequently, depending on the environment in which the conversational agent is deployed, this could serve as a criterion for selecting the agent with relatively better performance. It is important to note that the primary objective of `DialSim` is not to evaluate the inference speed of LLMs, but rather to assess the end-to-end performance of conversational agents, where techniques like model sharding and tensor parallelism can be a part of the conversational agent to decrease the response latency if needed.

To control the environmental factors that could affect time, we conducted all experiments under the same conditions as described in Appendix F.1.1. The rationale for setting a 6-second time limit in our experiments is detailed in Appendix F.1.2, and an analysis of the Internet speed for API-based models can be found in Appendix F.1.3.

## F.1.1 ENVIRONMENT CONTROL

Our simulator operates in real-time, requiring precise control of the experimental environment. Therefore, we conducted all experiments using the same hardware: NVIDIA RTX A6000 GPUs and an AMD EPYC 7702 64-Core Processor. To maintain consistent CPU performance, we allocated 10 cores for each experiment and ensured that no other processes were running simultaneously.

### F.1.2 AVERAGE TIME INTERVAL BETWEEN UTTERANCES

Each episode includes around 240 utterances and lasts about 18 minutes without commercial breaks. This means each utterance should occur roughly every 4.5 seconds. However, because the experiments used the A6000, which is slower than the latest hardware like the A100 or H100, we extended the interval to 6 seconds.

### F.1.3 INTERNET SPEED

The performance of API-based models can be affected by internet speed. To analyze this, we conducted a comparative analysis of the response times between API-based models and open-source models. In our analysis of agents using OpenAI Embedding-Session Sum., we found that the API-based agents achieved average response times of 1.50 seconds for ChatGPT-4o-mini, 1.73 seconds for ChatGPT-3.5 and 2.69 seconds for Gemini 1.0 pro. In comparison, agents using open-source models showed average response times ranging from 2.06 seconds (Gemma 2B) to 7.15 seconds (Tulu2 70B). These results suggest that, even when accounting for both internet communication and model inference, remote API-based models are generally faster than open-source alternatives. This indicates that internet latency has a minimal impact on our evaluation.

### F.2 QUESTION FORMAT

`DialSim` is a dataset that includes pairs of questions, answers, and choices. The questions are available in three formats: template-based multiple-choice, natural language multiple-choice, and open-ended. Users can choose any of these formats to evaluate the agent's performance.

First, we provide multiple-choice questions in both template and natural language formats. For example, a template-based question might be, "Who was going out with Paul in September 1994?" with choices "(A) Emily, (B) Monica, (C) Ryan, (D) Rachel, (E) I don't know". In contrast, the same question in natural language format could be phrased as, "Who was going out with Paul in September 1994? Was it Emily, Monica, Ryan, Rachel, or do you not know?"

Additionally, we offer the option to ask questions in an open-ended format (*e.g.*, "Who was going out with Paul in September 1994?") without providing answer choices. This approach allows us to evaluate the agent's ability to generate open-ended responses. The open-ended format is particularly useful for fan quiz-based questions, where some answers may require longer responses (*e.g.*, Question: "Why did Monica and Chandler say they were late getting to the hospital?" Correct answer: "Monica went back for her jacket").

For natural language multiple-choice and open-ended questions, a response is considered correct if it exactly matches the correct answer. If the response does not match exactly, the score is determined by comparing the response with the correct answer using a different language model (*i.e.*, GPT-4o mini).

### F.2.1 CHOICES IN MULTIPLE-CHOICE QUESTIONS

The number of questions based on fan quizzes was significantly smaller than the questions based on the TKG. Thus, 30% of the questions were intentionally extracted from the fan quiz-based during the simulation. Since each question has five choices, unanswerable questions were set to comprise 20% of the total to fairly stratify the correct answers.

### F.3 NUMBER OF RETRIEVED DIALOGUE HISTORY

By default, agents retrieved up to 20 utterances, 10 entire sessions, and 15 session summaries, depending on the storing method, though some LLMs with shorter context lengths retrieved fewer histories accordingly.

## G   EXPERIMENTAL RESULTS FOR THE BIG BANG THEORY AND THE OFFICE

The experimental results for The Big Bang Theory and The Office are provided in Table 11 and Table 12, respectively.

Table 11: The performances of the agents on The Big Bang Theory dialogue in `DialSim` (time limit = 6 seconds). We conducted experiments three times and reported the accuracies and the standard deviations.

| Type | Model | Base LLM | RAG-based | | | | | |
| | | | BM25 | | | OpenAI Embedding | | |
| | | | Utterance | Session Entire | Session Sum. | Utterance | Session Entire | Session Sum. |
|---|---|---|---|---|---|---|---|---|
| API | ChatGPT-4o-mini | 22.68 (2.12)[†] | 19.77 (2.02) | **36.63 (1.82)** | 30.10 (1.44) | 29.54 (0.71) | 32.34 (0.58) | 35.72 (0.81) |
| | ChatGPT-3.5 | **32.49 (1.72)** | 25.32 (1.20) | 35.59 (2.12) | 33.86 (1.09) | 27.81 (0.40) | 32.97 (0.86) | **37.02 (1.13)** |
| | Gemini 1.0 pro | 3.49 (0.69) | 25.87 (1.23) | 30.72 (0.18) | **38.16 (1.25)** | 37.42 (0.68) | 32.09 (0.44) | 36.30 (0.32) |
| Open | Tulu2-70B | 0.62 (0.13) | 21.08 (0.70) | 18.95 (1.07) | 22.36 (0.65) | 34.64 (0.69) | 9.08 (1.00) | 20.22 (1.48) |
| | Tulu2-7B | 0.53 (0.18) | 15.58 (1.34) | 22.26 (0.53) | 29.99 (0.57) | 16.84 (2.13) | 21.48 (0.77) | 28.69 (1.15) |
| | Llama3.1-70B | 0.25 (0.07)[†] | 21.55 (0.93) | 0.15 (0.12) | 1.26 (0.31) | 34.21 (1.59) | 3.89 (1.05) | 14.89 (1.74) |
| | Llama3.1-8B | 21.30 (1.68)[†] | 12.80 (1.06) | 25.50 (0.16) | 18.56 (0.99) | 23.10 (2.69) | 25.48 (3.65) | 20.75 (1.58) |
| | Mixtral-8x7B | 1.95 (0.34) | 15.91 (0.71) | 34.52 (1.12) | 16.83 (1.60) | 17.45 (0.49) | **34.98 (0.99)** | 13.83 (2.18) |
| | Mistral-7B | 3.11 (0.21) | 24.69 (1.82) | 34.26 (0.60) | 32.17 (1.39) | 30.23 (0.62) | 33.36 (0.56) | 29.19 (1.54) |
| | Gemma-7B | 16.40 (0.74) | 21.40 (2.33) | 19.74 (2.45) | 16.67 (0.40) | 24.50 (1.87) | 20.22 (1.39) | 16.12 (0.52) |
| | Gemma-2B | 1.56 (0.06) | **28.94 (0.35)** | 26.12 (2.22) | 33.47 (1.41) | 27.92 (0.68) | 29.40 (1.79) | 34.86 (3.20) |

†: Both ChatGPT-4o-mini and Llama3.1 support up to 128k tokens, but we limited them to 8k tokens due to high costs and GPU VRAM limits, respectively.

Table 12: The performances of the agents on The Office dialogue in `DialSim` (time limit = 6 seconds). We conducted experiments three times and reported the accuracies and the standard deviations.

| Type | Model | Base LLM | RAG-based | | | | | |
| | | | BM25 | | | OpenAI Embedding | | |
| | | | Utterance | Session Entire | Session Sum. | Utterance | Session Entire | Session Sum. |
|---|---|---|---|---|---|---|---|---|
| API | ChatGPT-4o-mini | 28.48 (1.01)[†] | 29.44 (0.62) | 43.16 (1.37) | 35.92 (2.50) | 37.81 (0.30) | 40.91 (0.37) | 42.83 (1.12) |
| | ChatGPT-3.5 | **36.54 (0.32)** | **36.63 (0.57)** | 45.33 (1.00) | **40.93 (0.13)** | 42.49 (1.24) | 43.04 (0.82) | 45.18 (0.56) |
| | Gemini 1.0 pro | 2.42 (0.18) | 35.11 (0.50) | **48.90 (1.57)** | 40.91 (0.75) | 44.72 (0.19) | 46.63 (0.89) | **45.82 (0.97)** |
| Open | Tulu2-70B | 0.46 (0.09) | 22.33 (1.00) | 35.52 (0.89) | 23.49 (1.16) | 38.61 (1.02) | 43.49 (1.27) | 23.54 (0.52) |
| | Tulu2-7B | 0.32 (0.04) | 25.86 (0.54) | 27.95 (1.03) | 36.60 (2.11) | 22.13 (0.33) | 29.50 (0.56) | 35.51 (1.40) |
| | Llama3.1-70B | 0.19 (0.07)[†] | 29.21 (0.56) | 13.31 (0.94) | 21.32 (5.22) | **47.41 (0.93)** | **47.07 (1.32)** | 19.46 (1.73) |
| | Llama3.1-8B | 21.87 (0.60)[†] | 22.03 (0.32) | 37.94 (1.28) | 29.16 (1.80) | 27.76 (3.52) | 37.67 (1.70) | 26.67 (0.83) |
| | Mixtral-8x7B | 1.53 (0.41) | 19.63 (0.79) | 34.35 (1.19) | 16.07 (0.56) | 20.02 (0.44) | 30.44 (1.69) | 12.43 (1.04) |
| | Mistral-7B | 2.55 (0.09) | 30.65 (0.45) | 41.16 (1.26) | 35.67 (1.68) | 36.92 (2.13) | 42.71 (1.24) | 37.65 (2.42) |
| | Gemma-7B | 17.81 (0.86) | 21.58 (0.61) | 25.62 (0.02) | 12.20 (0.57) | 24.88 (0.93) | 24.38 (0.52) | 15.70 (0.43) |
| | Gemma-2B | 0.83 (0.16) | 29.71 (0.69) | 28.11 (1.14) | 34.63 (0.94) | 31.54 (0.65) | 30.31 (0.16) | 33.37 (0.27) |

†: Both ChatGPT-4o-mini and Llama3.1 support up to 128k tokens, but we limited them to 8k tokens due to high costs and GPU VRAM limits, respectively.

# H  EXPERIMENTAL RESULTS FOR DIFFERENT TIME LIMITS

The experimental results for different time limits are shown in Figure 4 and Figure 5. Figure 4 illustrates the performance over different time limits in the BM25-Session Entire setting, while Figure 5 displays the performance in the Oracle setting. Due to the high costs, time-based experiments with ChatGPT-4o, Gemini-1.5 Pro, and Claude-3 Opus were conducted exclusively in the Oracle setting. One key observation from the results is the performance of ChatGPT-3.5, ChatGPT-4o-mini, and ChatGPT-4o. These models demonstrated consistent performance with quick inference times, handling up to a 3-second limit in the BM25-Session Entire setting and up to a 1-second limit in the Oracle setting. Consequently, these models are optimal for tasks requiring real-time communication without delays.

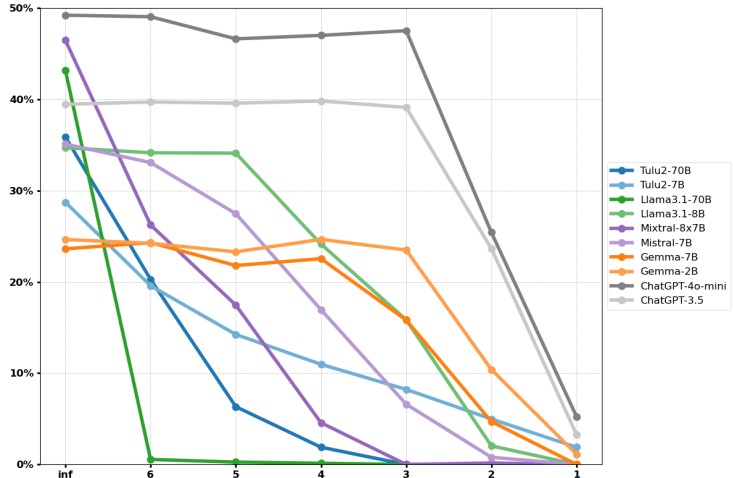

Figure 4: The experimental results for different time limits in the BM25-Session Entire setting.

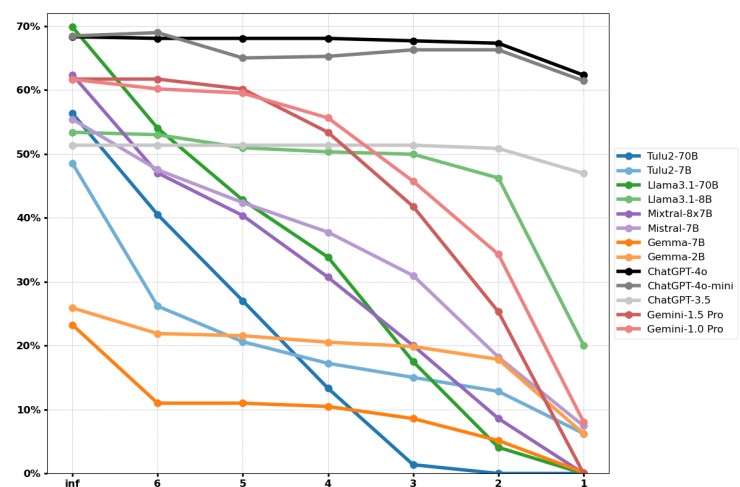

Figure 5: The experimental results for different time limits in the Oracle setting.

## I EXPERIMENTAL RESULTS IN THE ORACLE SETTING

Figure 6 shows the performance comparison between the BM25-Session Entire setting and the Oracle setting. These experiments were conducted without a time limit. Llama3.1-70B achieved the highest performance with a score of 69.86% in the Oracle setting.

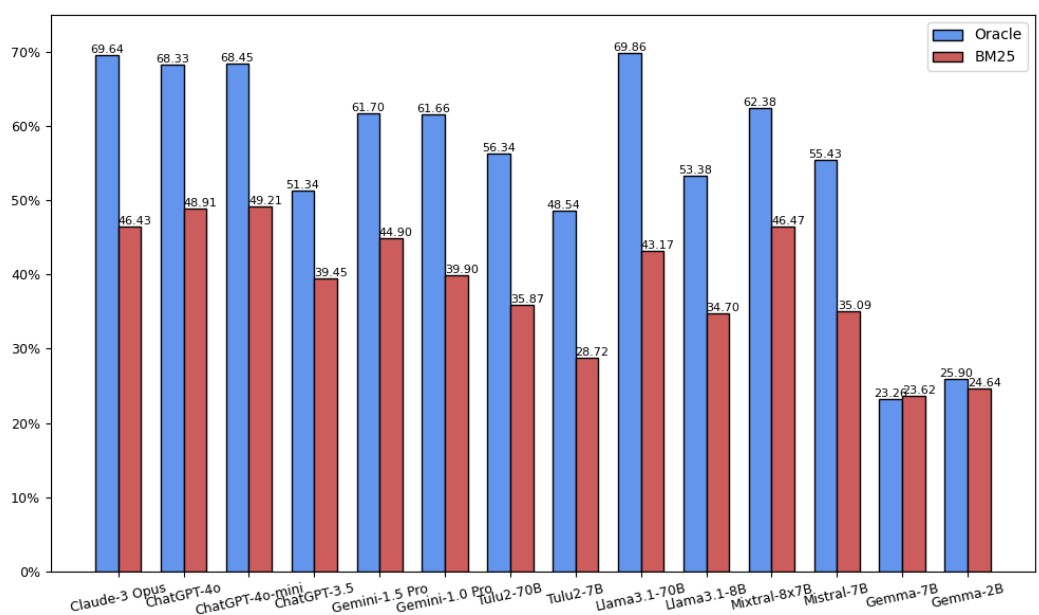

Figure 6: The performance comparison between the BM25-Session Entire setting and the Oracle setting.

## J EXPERIMENTAL RESULTS ON ADVERSARIAL TEST

In the adversarial test, we altered the characters' names and ran experiments under different conditions. Table 13 displays the results when characters' names were mixed with a 6-second time limit, while Table 14 shows the results without a time limit. Table 15 presents the results of changing characters' names to new ones with a 6-second time limit, while Table 16 shows the results without a time limit.

Table 13: The performances of the agents on Friends dialogue in DialSim (time limit = 6 seconds, with shuffled names). We conducted experiments three times and reported the accuracies and the standard deviations.

| | | | RAG-based | | | | | |
| | | | BM25 | | | OpenAI Embedding | | |
| Type | Model | Base LLM | Utterance | Session Entire | Session Sum. | Utterance | Session Entire | Session Sum. |
|---|---|---|---|---|---|---|---|---|
| Open | Tulu2-70B | 0.31 (0.13) | 17.03 (0.94) | 15.20 (0.87) | 18.45 (1.04) | 26.89 (0.54) | 6.92 (0.47) | 12.86 (1.23) |
| | Tulu2-7B | 0.73 (0.29) | 12.50 (1.73) | 17.58 (1.14) | 24.21 (1.09) | 10.20 (0.21) | 14.26 (0.92) | 21.03 (0.58) |
| | Llama3.1-70B | 0.51 (0.00)[†] | **27.84 (1.89)** | 0.60 (0.06) | 13.84 (2.31) | **35.67 (1.89)** | 1.53 (028) | 20.90 (0.37) |
| | Llama3.1-8B | **25.84 (1.16)**[†] | 25.24 (0.30) | **28.86 (1.01)** | 24.56 (0.99) | 28.86 (1.10) | **32.35 (1.51)** | 24.05 (1.36) |
| | Mixtral-8x7B | 1.77 (0.19) | 14.03 (0.12) | 21.11 (1.07) | 15.50 (0.68) | 13.14 (0.83) | 18.03 (0.44) | 18.47 (0.55) |
| | Mistral-7B | 2.34 (0.17) | 22.29 (1.43) | 27.08 (0.99) | 24.15 (1.76) | 25.17 (1.74) | 26.76 (2.64) | 23.81 (2.53) |
| | Gemma-7B | 18.87 (1.43) | 22.85 (0.81) | 22.96 (1.34) | 17.95 (0.62) | 25.46 (2.08) | 21.53 (1.00) | 17.66 (1.31) |
| | Gemma-2B | 0.78 (0.22) | 22.99 (0.66) | 25.48 (1.54) | **25.86 (2.48)** | 25.08 (1.34) | 25.21 (0.22) | **26.14 (1.71)** |

†: Llama3.1 supports up to 128k tokens, but we limited it to 8k tokens due to GPU VRAM limits.

Table 14: The performances of the agents on Friends dialogue in `DialSim` (without a time limit and with shuffled names). We conducted experiments three times and reported the accuracies and the standard deviations.

| Type | Model | Base LLM | RAG-based | | | | | |
|---|---|---|---|---|---|---|---|---|
| | | | BM25 | | | OpenAI Embedding | | |
| | | | Utterance | Session Entire | Session Sum. | Utterance | Session Entire | Session Sum. |
| Open | Tulu2-70B | 2.54 (0.21) | 26.47 (1.91) | 31.75 (1.71) | 30.94 (2.41) | 31.90 (1.25) | 29.83 (1.03) | 31.86 (2.13) |
| | Tulu2-7B | 1.15 (0.06) | 28.20 (1.63) | 27.64 (2.37) | 27.78 (1.32) | 28.98 (0.96) | 25.03 (0.94) | 29.08 (2.47) |
| | Llama3.1-70B | 31.38 (1.01)[†] | 29.08 (1.57) | 36.48 (2.51) | **36.91 (0.36)** | **35.89 (0.65)** | 39.80 (1.42) | **39.00 (0.87)** |
| | Llama3.1-8B | 27.16 (1.62)[†] | 25.76 (1.42) | 30.61 (1.25) | 29.59 (1.25) | 30.91 (0.99) | 29.76 (1.26) | 31.59 (0.69) |
| | Mixtral-8x7B | **34.19 (0.68)** | 25.23 (1.19) | **37.72 (0.96)** | 29.48 (0.87) | 29.09 (1.46) | 31.78 (1.71) | 29.45 (0.04) |
| | Mistral-7B | 27.78 (1.62) | 25.02 (1.26) | 30.65 (1.39) | 24.99 (1.51) | 27.34 (0.49) | 27.97 (1.31) | 26.97 (1.45) |
| | Gemma-7B | 17.98 (2.15) | 21.64 (0.39) | 22.31 (2.15) | 18.66 (1.55) | 25.97 (1.92) | 21.79 (0.40) | 21.22 (0.59) |
| | Gemma-2B | 1.04 (0.19) | 24.19 (0.82) | 25.25 (1.02) | 24.32 (1.55) | 25.03 (0.66) | 25.44 (1.96) | 23.62 (0.36) |

†: Llama3.1 supports up to 128k tokens, but we limited it to 8k tokens due to GPU VRAM limits.

Table 15: The performances of the agents on Friends dialogue in `DialSim` (time limit = 6 seconds, with new names replaced). We conducted experiments three times and reported the accuracies and the standard deviations.

| Type | Model | Base LLM | RAG-based | | | | | |
|---|---|---|---|---|---|---|---|---|
| | | | BM25 | | | OpenAI Embedding | | |
| | | | Utterance | Session Entire | Session Sum. | Utterance | Session Entire | Session Sum. |
| Open | Tulu2-70B | 0.21 (0.07) | 18.24 (0.84) | 20.60 (1.00) | 18.64 (1.81) | 31.71 (2.22) | 7.82 (1.57) | 17.31 (0.61) |
| | Tulu2-7B | 0.74 (0.15) | 13.19 (0.35) | 19.54 (1.29) | 26.07 (2.00) | 13.87 (0.71) | 18.35 (1.21) | **27.48 (2.04)** |
| | Llama3.1-70B | 0.64 (0.10)[†] | **29.29 (0.59)** | 0.60 (0.12) | 15.07 (5.12) | **39.08 (0.99)** | 2.43 (0.10) | 18.18 (0.16) |
| | Llama3.1-8B | **26.61 (1.24)**[†] | 26.86 (0.78) | 31.20 (1.87) | **27.08 (0.63)** | 24.82 (1.07) | **31.72 (1.66)** | 22.69 (0.63) |
| | Mixtral-8x7B | 2.41 (0.40) | 14.90 (0.82) | 23.55 (0.40) | 15.64 (0.47) | 16.43 (1.68) | 22.95 (0.68) | 13.22 (1.61) |
| | Mistral-7B | 3.35 (0.58) | 24.44 (1.13) | **31.39 (0.70)** | 24.26 (1.60) | 29.82 (0.95) | 30.21 (0.90) | 23.90 (0.51) |
| | Gemma-7B | 18.05 (0.97) | 22.52 (0.81) | 20.64 (0.26) | 16.63 (1.59) | 23.41 (1.26) | 18.34 (0.82) | 19.48 (2.45) |
| | Gemma-2B | 0.47 (0.13) | 24.31 (0.96) | 24.77 (0.74) | 25.74 (1.46) | 28.41 (1.20) | 24.68 (1.45) | 24.75 (1.50) |

†: Llama3.1 supports up to 128k tokens, but we limited it to 8k tokens due to GPU VRAM limits.

Table 16: The performances of the agents on Friends dialogue in `DialSim` (without a time limit and with new names replaced). We conducted experiments three times and reported the accuracies and the standard deviations.

| Type | Model | Base LLM | RAG-based | | | | | |
|---|---|---|---|---|---|---|---|---|
| | | | BM25 | | | OpenAI Embedding | | |
| | | | Utterance | Session Entire | Session Sum. | Utterance | Session Entire | Session Sum. |
| Open | Tulu2-70B | 2.17 (0.46) | 27.24 (1.17) | 33.34 (1.17) | 32.85 (1.85) | 34.95 (0.47) | 29.41 (1.22) | 33.55 (2.79) |
| | Tulu2-7B | 0.63 (0.26) | **30.26 (1.03)** | 27.68 (1.24) | 30.98 (1.08) | 30.99 (0.22) | 27.93 (1.97) | 31.80 (2.05) |
| | Llama3.1-70B | 31.03 (1.91)[†] | 28.91 (2.33) | 38.44 (5.98) | **41.68 (3.68)** | **38.40 (1.10)** | 40.83 (1.07) | **44.27 (0.57)** |
| | Llama3.1-8B | 26.65 (1.19)[†] | 25.80 (0.18) | 32.01 (1.10) | 30.48 (1.30) | 29.50 (1.10) | 32.82 (0.63) | 32.23 (2.32) |
| | Mixtral-8x7B | **38.92 (1.61)** | 26.91 (1.46) | **39.98 (2.98)** | 27.02 (0.42) | 30.27 (1.37) | 35.89 (0.37) | 29.52 (1.28) |
| | Mistral-7B | 29.10 (1.34) | 23.33 (0.83) | 34.59 (0.80) | 27.87 (2.66) | 30.59 (2.09) | 30.45 (0.89) | 27.93 (0.99) |
| | Gemma-7B | 17.37 (0.77) | 22.58 (1.62) | 21.41 (1.53) | 21.61 (1.53) | 23.90 (0.90) | 21.61 (1.09) | 20.88 (0.91) |
| | Gemma-2B | 0.37 (0.07) | 23.48 (1.85) | 23.13 (1.14) | 25.72 (2.67) | 29.12 (1.90) | 24.88 (1.40) | 24.81 (1.31) |

†: Llama3.1 supports up to 128k tokens, but we limited it to 8k tokens due to GPU VRAM limits.

# K   LLM's Prior Knowledge of the TV shows

We asked ChatGPT-4o to explain the plot of specific episodes of Friends. It accurately described the plots, as shown in Figure 7, 8. Notably, it provided these answers without any web browsing, suggesting that ChatGPT-4o might have learned about these TV shows during its pre-training process.

Tell me the plot of episode 7 from season 2 of the TV show Friends.

Sure! Episode 7 of Season 2 of "Friends" is titled "The One Where Ross Finds Out."

Here's the plot summary:

Rachel is on a date with Michael, a man she recently met, but she is unable to focus on him because she is still in love with Ross. Meanwhile, Ross is still dating Julie, which complicates Rachel's feelings even further. During the date, Rachel gets drunk and leaves a message on Ross's answering machine, confessing her love for him.

The next day, Ross hears the message and confronts Rachel at Central Perk. They have an intense conversation where Rachel tries to play it off, but Ross presses on, leading to a passionate kiss. Ross is torn between his feelings for Rachel and his current relationship with Julie.

In a subplot, Chandler is trying to quit smoking and resorts to a hypnosis tape to help him stop. However, the tape is designed for women, which leads to Chandler inadvertently picking up some feminine habits.

Monica is helping Chandler with his attempts to quit smoking, and Joey and Phoebe provide comedic relief with their antics. The episode ends on a cliffhanger, leaving viewers wondering what Ross will decide regarding his relationship with Rachel and Julie.

Figure 7: The result of asking ChatGPT-4o to explain Season 2, Episode 7 of Friends.

Tell me the plot of episode 14 from season 3 of the TV show Friends.

Episode 14 of season 3 of the TV show "Friends" is titled "The One with Phoebe's Ex-Partner." Here is a summary of the plot:

Phoebe's former singing partner, Leslie, shows up and wants to get back together. Initially, Phoebe is excited about the reunion, but their rekindled partnership quickly hits a snag when Leslie sells out by using "Smelly Cat" in a commercial. This betrayal causes Phoebe to sever ties with Leslie once again.

Meanwhile, Chandler dates a woman named Ginger who has a wooden leg, a fact he only discovers after Ross inadvertently tells him. This revelation makes Chandler uncomfortable, leading to some awkward and humorous situations. Joey also remembers Ginger from a previous encounter where he accidentally threw her artificial leg into a fire.

Ross is dealing with relationship issues with Rachel. They have a fight over a list Ross made, comparing her to Julie, his previous girlfriend. Despite this, Ross tries to make amends by taking her to a planetarium. However, their attempt at romance is interrupted by a loud child in the audience, and Rachel's feelings remain unresolved.

The episode balances the comedic elements with more serious moments, showcasing the dynamics between the friends and their relationships.

Figure 8: The result of asking ChatGPT-4o to explain Season 3, Episode 14 of Friends.

## L ANNOTATOR INSTRUCTIONS

Figure 9 and Figure 10 show the screenshots of the dataset labeling process. Figure 9 illustrates the annotation process for the questions based on fan quizzes, and Figure 10 describes the review process for selecting triples for the TKG.

```
#########
[OPEN the 'Pilot' episode script]
#########

Do you want to do labeling? y/n
█
```

Question Revision

```
Question ID is  45

[Original Question]: 1."TOW Monica Gets a Roommate" - Who did Monica have a date with?
[New Question]: Who was Monica's date with?

1. Has the [Original Question] been successfully rewritten into [New Question]? y/n
y
```

Determine Answerability

```
Correct Answer of the Question: Paul
GPT-4 said that this question is answerable.
2. Is the given question answerable? y/n
y
```

Evidence Scene Revision

```
Correct Answer of the Question: Paul
GPT-4 said that we can answer the question using [1, 5, 8, 11, 14] scenes.
3. Are the given scenes enough? y/n/r
█
```

Figure 9: The actual process of annotating questions from fan quizzes.

```
#########
[OPEN the 'Monica Gets A Roommate' episode script]
#########

Scene number list is  [1]

0 ['phoebe', 'ex-boyfriend', 'carl']
1 ['ross', 'ex-spouse', 'carol']
2 ['monica', 'brother', 'ross']
3 ['rachel', 'fiance', 'barry']
4 ['rachel', 'ex-fiance', 'barry']
5 ['monica', 'friend', 'rachel']
6 ['rachel', 'visited place', "monica's building"]
7 ['rachel', 'hometown', 'the city']

Which triples can be survived? (e.g. int, int, int)
█
```

Figure 10: The actual process of reviewing extracted triples.

