# OpenReview forum: "DialSim: A Real-Time Simulator for Evaluating Long-Term Multi-Party Dialogue Understanding of Conversational Agents"
_ICLR.cc/2025/Conference — ICLR 2025 Conference Withdrawn Submission_

### Official Review · Reviewer_4WJV · 2024-11-01

**Soundness:** 3
**Presentation:** 2
**Contribution:** 3
**Rating:** 6
**Confidence:** 3

**Summary:**

This paper introduces DialSim, a dialogue simulator for real-time evaluation of a conversational agent’s long-term multi-party dialogue understanding. It provides time-constrained evaluation with high-quality and extensive long-term dialogue.

**Strengths:**

1.The paper is well-motivated. It introduces real-time constraints in evaluating conversational agents, which is significant in real-world scenarios.

2.Although the task is synthetic, the process of test case generation is novel and reasonable. It also provides flexibility in both generating questions and generating test cases.

3.The experiments encompass RAG-based methods and analysis on errors, storing history and time limit, which are comprehensive and in depth.

**Weaknesses:**

1.Although DialSim simulates long-term dialogues averaging 350k tokens, the experiment shows that ChatGPT-4o-mini and Llama-3.1 still give correct answer frequently with 8k tokens context length, which means it does not require long context capabilities to solve this task.

2.Data contamination might be a problem. Since DialSim uses scripts from popular TV shows, conversational agents might know the answer of question based on their own knowledge rather than the dialogue.

3.Limited variety of task types and data sources. The paper only focuses on question-answering and TV shows scripts, which may restrict the scope of the benchmark in thoroughly evaluating capabilities of conversational agents.

**Questions:**

1.Could you provide an analysis on text length utilization and data contamination? I believe it is important to evaluate the quality of this benchmark.

2.I am curious about the performances of LLMs on unanswerable questions, since they demonstrate the extent to which LLMs understand the entire dialogue.

---

> ### Author Response · Authors · 2024-11-14
>
> Thank you for reviewing our paper and providing constructive feedback. To address any potential misunderstandings you might have, we believe it would be helpful to discuss each of the highlighted weaknesses individually. The authors are committed to monitoring comments around the clock during the remaining two-week discussion period and are prepared to engage in further discussion to clarify any points as needed.
>
> ``W1. Although DialSim simulates long-term dialogues averaging 350k tokens, the experiment shows that ChatGPT-4o-mini and Llama-3.1 still give correct answer frequently with 8k tokens context length, which means it does not require long context capabilities to solve this task.``
>
> First, ChatGPT-4o-mini and Llama-3.1 70B both recorded less than 40% accuracy which is far below the oracle performance (69.89%, line 445), and 10% lower than the maximum performance achievable by RAG-based methods. Second, we ran our simulator to measure the length of the history between the evidence scene for an answerable question and the point at which the question was asked. The average length was measured to be 160k tokens.
>
> Do these two points resolve your misunderstanding (i.e. DialSim does not require long context capabilities)?

---

> > ### Comment · Reviewer_4WJV · 2024-11-19
> >
> > I am still somewhat unclear about the performance. The ChatGPT-4o-mini model achieves nearly 40% accuracy with an 8k context window, which suggests the following:
> >
> > 1.The performance of ChatGPT-4o-mini is far above random guessing (20%), indicating it can solve many problems within its 8k context window.
> >
> > 2.Given that the oracle performance is close to 70%, only about 30% of the questions require context beyond the 8k token limit to be answered accurately.
> >
> > Based on this and considering the average length of the history is 160k, I expect a more in-depth analysis and detailed clarification of this issue.

---

> > > ### Author Response · Authors · 2024-11-20
> > >
> > > Thank you for actively participating in the discussion and for providing such valuable feedback.
> > >
> > > Before addressing this concern directly, we would like to first provide an answer to Question 2: ``I am curious about the performances of LLMs on unanswerable questions, since they demonstrate the extent to which LLMs understand the entire dialogue.`` This context will help us better address your concern.
> > >
> > > When designing the conversational agent, we explicitly included a prompt instruction for the agent to respond with "I don't know" when it lacks sufficient evidence to answer a question (as detailed in Appendix A), so that it could handle not only answerable questions but also unanswerable questions. Consequently, the agent frequently responded with "I don't know" to questions it was unsure about, leading to high accuracy on unanswerable questions. Notably, the top-performing agents—GPT-4o-mini / Session-Entire / BM25 and Llama-3.1 70B / Session-Sum / OpenAI emb—achieved nearly 90% accuracy on unanswerable questions (without time constraints). However, given their overall accuracy of approximately 49%, this implies they achieved around 40% accuracy on answerable questions.
> > >
> > > This high accuracy on unanswerable questions is also observed in the Base LLM setting of GPT-4o-mini and Llama-3.1 70B. For instance, GPT-4o-mini achieved an accuracy of 92.09% on unanswerable questions, but its performance dropped significantly to just 24.91% for answerable ones. In contrast, in the BM25-Session Entire setting, it achieved an accuracy of 39.66% on answerable questions. Therefore, the accuracy gap for answerable questions highlights the challenges posed by the need for long-term dialogue understanding.
> > >
> > > Based on the above explanation, we hope we have addressed your concerns adequately. We plan to incorporate this analysis into our paper. Your insights have significantly contributed to improving our work, and we greatly appreciate them. If any part of our intent or explanation has been misunderstood, please feel free to share further details, and we will gladly clarify.

---

> > > > ### Comment · Reviewer_4WJV · 2024-11-24
> > > >
> > > > Thanks for the explanation and providing additional results. I will raise my score accodingly.

---

> > > > > ### Author Response · Authors · 2024-11-24
> > > > >
> > > > > Thank you for actively participating in the discussion! With this, we have completed the discussion regarding Weakness 1 and Question 2 that you pointed out. We would now like to continue the discussion regarding Weakness 2 that you highlighted:
> > > > >
> > > > > ```2. Data contamination might be a problem. Since DialSim uses scripts from popular TV shows, conversational agents might know the answers to questions based on their own knowledge rather than the dialogue.```
> > > > >
> > > > > Naturally, we also considered that the LLM has likely already learned from TV show scripts (see Appendix K). Therefore, we have proposed two methods to mitigate this issue. First, as indicated in lines 288-293 and 311-316 of the paper, our dataset includes unanswerable questions—questions that the agent cannot answer based solely on the dialogue provided up to that point. This allows for a more accurate evaluation by ensuring that memorization would lead to incorrect answers. Second, we implemented adversarial tests by changing/shuffling the names of characters in the dialogue, thus providing benchmark users an option to test how much their conversational agent (i.e. LLM) is relying on prior knowledge.
> > > > >
> > > > > With these two methods, we aim to prevent data contamination problem and ensure a fair evaluation of the agents.
> > > > >
> > > > > Does our response address your concern? If there is still something unclear or logically inconsistent, we would greatly appreciate it if you could elaborate further so we can provide a more detailed response.

---

> > > > > > ### Comment · Reviewer_4WJV · 2024-12-02
> > > > > >
> > > > > > Thank you again for your response which addresses my concern on data contamination of this benchmark. I have raised my score and will maintain my positive score.

---

### Official Review · Reviewer_3tT9 · 2024-11-04

**Soundness:** 2
**Presentation:** 2
**Contribution:** 2
**Rating:** 3
**Confidence:** 4

**Summary:**

This paper proposed DialSim, a real-time multi-party dialogue simulator. It is designed with a time limit, random QA, to fully evaluate the conversational LLM agents. Evaluation experiments are conducted on a wide range of baseline LLMs.

**Strengths:**

1. The authors make a great job to provide a high-quality and large-scale dialogue dataset from TV shows. The conversations in the data are multi-party, a promising and useful domain in the community.

2. The method used to construct the data is carefully-designed, to ensure the quality, e.g. using temporal knowledge graph, character style rephrasing.

**Weaknesses:**

1. In spite of the construction of the data, the work has minor contribution. While the evaluation spans a wide range of models, the conclusions are normal. I find the adversarial test in the paper interesting, where the model gets rid of the prior knowledge when answering the question, but the authors decide to put them in Appendix. There is space of one more page.

2. Since this work releases a high-quality dataset, it can be very helpful to demonstrate more concrete examples of the dialogues and conduct comparison to previous ones. However, there is no illustrative examples in the paper. It is hard for readers to justify the quality of the data as well as the effectiveness of the generating methods.

3. It is not clear how or to what extent, the mentioned data generation methods e.g. using knowledge graph, can facilitate the quality of the data. The paper lacks corresponding endeavors to illustrate this.

4. The work focuses on multi-party dialogue, a challenging task even for recent LLMs. However, it is a pity that the relevant discussion cannot be found in the related work or analysis of the paper.

5. The constructed DialSim is mainly characterized by its time limit and scale, i.e. some regular factors rather than some other promising factors. Therefore, the contribution of this paper is not positive considering the recent advance in the AI community.

**Questions:**

N/A

---

> ### Author Response · Authors · 2024-11-14
>
> Thank you for reviewing our paper and providing constructive feedback. To address any potential misunderstandings you may have, we propose discussing each of the points noted as weaknesses, one by one. The authors are fully committed to monitoring comments around the clock during the remaining two-week discussion period and are prepared to clarify any areas that may require further explanation. We look forward to resolving any concerns through this discussion.
>
> ``W3. It is not clear how or to what extent, the mentioned data generation methods e.g. using knowledge graph, can facilitate the quality of the data. The paper lacks corresponding endeavors to illustrate this.``
>
> Given the explanation, figures and examples in Section 3.2 regarding the dataset construction process, could you explain why you find “the paper lacks corresponding endeavors”?

---

> > ### Comment · Reviewer_3tT9 · 2024-11-15
> >
> > Sure. My main concern is whether the authors can justify the effectiveness of the proposed novel methods, such as TKG, used during the dataset construction process. For instance, how does TKG contribute to improving data quality? In the submission, I cannot find evidence or experimental results to prove this. Without such justification, it is hard for me to evaluate the novelty and usefulness of the methods.
> >
> > I could miss something important in the paper. If such evidence or results have been provided, please indicate which part of the paper they can be found.
> >
> > Sure. I am happy to adjust the rating according to the discussion.

---

> > > ### Author Response · Authors · 2024-11-15
> > >
> > > Thank you very much for dedicating your time to actively engaging in the discussion about our paper.
> > >
> > > We employed two methods for question generation: a fan quiz-based approach and a temporal knowledge graph (TKG)-based approach.
> > >
> > > First, as outlined in Section 3.2.2, we utilized a fan quiz website, FunTrivia, to generate our questions. Fan quizzes cover a range of difficulty levels and emphasize major events from each episode, making them suitable for evaluating dialogue comprehension. To filter questions suitable for our task and map these questions to the corresponding evidence scenes, we initially employed ChatGPT-4 and then manually reviewed the results.
> > >
> > > Additionally, as described in Section 3.2.3, while fan quizzes are useful for question generation, they are episode-specific and user-generated. This means the questions do not span multiple episodes, and their quantity is limited (around 1K). To address this limitation, we constructed a knowledge graph for each session and used it to generate additional questions. As shown in Table 2, the TKG-based approach allows for a significantly larger number of questions compared to the fan quiz-based method. As for the idea of TKG itself, we can't claim that it is entirely our own novel idea, as we've borrowed the KG schema from previous work [1]. However, using TKG to generate multi-hop QA pairs across long-term dialogues can be seen as one of our work's contributions.
> > >
> > > In our experiments, as noted in Line 450 of Section 4.3, agents tend to score lower on TKG-based questions (46.42%) than on those derived from fan quizzes (58.80%). This indicates that TKG-based questions are more challenging than fan quiz-based questions. Among the TKG-based questions, the accuracy for one-hop questions was 66.67%, whereas two-hop questions had a significantly lower accuracy of 13.53%, highlighting the greater difficulty of two-hop questions.
> > >
> > > Based on the above explanation, we hope we have addressed your concerns adequately. If any part of our intent or explanation has been misunderstood, please feel free to share further details, and we will gladly clarify.
> > >
> > > [1] Dian Yu, Kai Sun, Claire Cardie, and Dong Yu. Dialogue-based relation extraction. In Dan Jurafsky, Joyce Chai, Natalie Schluter, and Joel Tetreault (eds.), Proceedings of the 58th Annual Meeting of the Association for Computational Linguistics, pp. 4927–4940, Online, July 2020. Association for Computational Linguistics. doi: 10.18653/v1/2020.acl-main.444. URL https://aclanthology.org/2020.acl-main.444.

---

> ### Author Response · Authors · 2024-12-04
>
> The period for the reviewer to change the score is only a few hours away from ending. It is quite frustrating that a score of 3 could be finalized without even a discussion. We kindly ask you to fulfill your responsibility as a reviewer.

---

### Official Review · Reviewer_La8u · 2024-11-04

**Soundness:** 4
**Presentation:** 4
**Contribution:** 3
**Rating:** 8
**Confidence:** 4

**Summary:**

The paper introduces DialSim, a real-time dialogue simulator, and builds a dataset using this simulator, to evaluate the LLM's ability to play as the role of a character from popular TV shows, and correctly answer randomized questions related to the show w/ or wo/ history context, within a given time limit.

The paper presents details about the dataset construction, and compares the datasets with several existing related works. The dataset features its large scale, multiple characters, long context that spans several years, relationships between characters that evolve over time, etc.

The paper conducts extensive experiments on the datasets, with closed- and open-source SOTA models. Results are presented to explore (1) w/ and wo/ history context (2) different forms of history contexts (3) different retrieval methods (4) w/ and wo/ time limit. Then, the paper analyzes the results in detail and provides useful insights.

**Strengths:**

1. The paper presents a highly novel method to simulate a conversational dataset that can evaluate LLM's abilities from a very practical and real-world scenario consideration (time budget, complex conversation setting, long history, etc.)

2. The paper conducts lots of experiments to compare a list of SOTA models given different settings. The results show that current models cannot handle the task well, demonstrating the effectiveness of the task and dataset.

3. The paper organization and clarity are pretty good.

**Weaknesses:**

1. The paper does not clearly distinguish between (1) a simulator that helps to generate multi-choice QA / open QA given TV show scripts, and (2) a dataset built with the simulator. This causes some confusions. For example, I wonder:

1) For the evaluation, are the experiments on different datasets conducted on a same, fixed dataset? Does the evaluation incorporate any randomness?

2) If there is a fixed evaluation dataset, then what are its statistics? I noticed that Table 2 presents "Average Fan Quiz Questions per Session=56.7", is this the number of question candidates, or number of actual selected questions?

2. As the paper also points out, although it considers real-world scenarios like time budget and long history, the data source of TV show scripts may limit its application to real-world applications.

**Questions:**

As written in the Weaknesses section, I'd like to know if there is a fixed evaluation dataset for the evaluation. If so, please present more details about this dataset.

---

> ### Author Response · Authors · 2024-11-14
>
> Thank you very much for reviewing our paper and providing constructive feedback. To address any potential misunderstandings you might have, we believe it would be helpful to discuss each of the highlighted weaknesses individually. The authors are committed to checking comments around the clock throughout the remaining two-week discussion period, and we are fully prepared to clarify any points that may require further explanation through ongoing discussion.
>
> ``W1. The paper does not clearly distinguish between (1) a simulator that helps to generate multi-choice QA / open QA given TV show scripts, and (2) a dataset built with the simulator. This causes some confusions.``
>
> Before conducting the simulation, we first generated question candidates (fixed dataset) for each session in the dialogue. Then, using these question candidates, we proceed with the simulation to evaluate the agent's performance.
>
> To explain in more detail, we created question candidates for each session by employing a fan quiz-based question generation method and a temporal knowledge graph-based question generation method. This process resulted in a fixed dataset that includes dialogue sessions and their corresponding question candidates, with dataset statistics provided in Table 2.
>
> After completing the dataset, as described in Section 3.1.2, once the simulation begins, as the dialogue progresses, a random question is selected from the fixed question candidates of each session and is asked to the agent at a random timing. Through this approach, we ensure a high level of randomness and emulate an unpredictable real-world environment.
>
> Could you let us know if our explanation helped clarify your concerns? If there are any areas that remain unresolved, please feel free to let us know.

---

### Official Review · Reviewer_rZVL · 2024-11-04

**Soundness:** 3
**Presentation:** 3
**Contribution:** 1
**Rating:** 3
**Confidence:** 4

**Summary:**

This paper introduces DialSim, a simulator for dialogue based on popular TV series. It leverages fan quizz websites and ChatGPT in order to generate realistic dialogues and Q&As. The introduced dataset is larger than available alternatives and supports long-range dialog and multi-hop questions. Multiple LLMs are tested against this simulator, showing a direct comparison between mainstream open-weight models and available APIs.

**Strengths:**

# Originality

The domain covered by the paper has been covered by multiple works, however the scale of the proposed dataset is higher than available alternatives. The paper provides a good coverage of the existing literature and its coverage. The method proposed to generate long-range dialogues by leveraging a temporal knowledge graph is novel, and as a whole, the data pipeline leverages existing LLMs in an interesting way.


# Quality and clarity

The paper is well-written, it presents very clearly the various steps of the data creation process and the setup in which mainstream LLMs were tested.


# Significance

The significance of this paper is not clear to me. While extracting long-term dialogue data for benchmarking is broadly useful to NLP researchers, the fact that this data comes from extremely popular sources makes it very hard to claim actual scene understanding by the model. (see weaknesses section)

**Weaknesses:**

## Time limit scenario
The paper puts a time limit constraint on the responses under some hardware capacity constraints. However, it presents both open-weight LLMs and available APIs under the same light, although the hardware used by APIs is unknown, and is likely to be significantly more powerful than A6000 GPUs.

## Ablating the data preprocessing pipeline
The data creation involves creating prompting of LLMs in order to filter questions and apply personal style transfer. It also leverages a temporal knowledge graph in order to formulate complex, multi-hop questions. It would have made sense to justify this step and show how the current LLMs cannot efficiently solve this task.

## Data leakage is probably significant, making the paper's relevance questionable

Most importantly, this paper introduces a dataset based on highly popular sources. Any current LLM has likely been trained on fan quizzes, episode synopses or even full dialogue scripts for these TV shows. As such, it is impossible to attribute correct answering of the model to an actual understanding of the dialogue as opposed to correct memorization of its training set. The paper attempts to mitigate this by running an "adversarial test", however it is likely that training set memorization goes beyond the memorization of mainstream character names. For instance, any question about how a character bought their new boots in a TV-show-like setup will be irremediably tainted by the memorization of the first episode of Friends. As a consequence, it is not clear to me how relevant the overall approach can be, as it actually relies on the mainstream popularity of the content in order to properly source and curate data.

**Questions:**

# How should practitioners approach data leakage with this benchmark?

As mentioned in the weaknesses section, it seems likely that the popularity of the content makes for very high data leakage in training sets.
Have you tested stronger "adversarial" testing methods than simply swapping character names? How can practitioners evaluate their dialogue agents in this framework while making sure that they're actually testing long-range dialogue understanding and not merely training set memorization?

---

> ### Author Response · Authors · 2024-11-14
>
> Thank you very much for reviewing our paper and for providing constructive feedback. We believe it would be helpful to address the points noted as weaknesses by discussing them individually to clarify any misunderstandings you might have. The authors are committed to actively monitoring comments around the clock throughout the remaining two-week discussion period. If any part requires further clarification, we are fully prepared to resolve it through discussion.
>
> ``W3: Data leakage is probably significant, making the paper's relevance questionable``
>
> ``Q1: How should practitioners approach data leakage with this benchmark?``
>
> Naturally, we also considered that the LLM has likely already learned from TV show scripts (see Appendix K). Therefore, we have proposed two methods to mitigate this issue. First, as indicated in lines 288-293 and 311-316 of the paper, our dataset includes unanswerable questions—questions that the agent cannot answer based solely on the dialogue provided up to that point. This allows for a more accurate evaluation by ensuring that memorization would lead to incorrect answers. Second, we implemented adversarial tests by changing/shuffling the names of characters in the dialogue, thus providing benchmark users an option to test how much their conversational agent (i.e. LLM) is relying on prior knowledge.
>
> Why would you think these two methods are not proper methods to handle data leakage?

---

> ### Author Response · Authors · 2024-12-04
>
> The period for the reviewer to change the score is only a few hours away from ending. It is quite frustrating that a score of 3 could be finalized without even a discussion. We kindly ask you to fulfill your responsibility as a reviewer.

---

### Author Response · Authors · 2024-11-19
**Sincerely expecting further discussions with Reviewers**

Dear Reviewers,

We deeply value your feedback and hope to understand which aspects of our paper need improvement or clarification. We kindly ask you to point out specific parts that failed to convince you or led to any misunderstanding.

Please know that we are fully committed to engaging in discussions throughout the review period and look forward to addressing your concerns.

---

### Author Response · Authors · 2024-11-22
**Discussion Period Nearing Its End**

Dear Reviewers,

Thank you once again for dedicating your time to reviewing our paper. As the discussion period is approaching its conclusion, we kindly request your participation in the discussion. Through this discussion, we hope to address any potential misunderstandings about our paper and to hear more detailed feedback on any areas where the paper may be lacking.

We remain fully committed to actively participating in these discussions throughout the review period and look forward to addressing any concerns you may have.

---

### Note · Authors · 2024-12-15

I have read and agree with the venue's withdrawal policy on behalf of myself and my co-authors.